# Understanding The Robustness of Self-supervised Learning Through Topic Modeling

**Zeping Luo**[*], **Shiyou Wu**[*], **Cindy Weng**[*], **Mo Zhou**[†], **Rong Ge**[†]
Duke University, USA
[*]{zeping.luo,shiyou.wu,cindy.weng}@duke.edu, [†]{mozhou,rongge}@cs.duke.edu

## Abstract

Self-supervised learning has significantly improved the performance of many NLP tasks. However, how can self-supervised learning discover useful representations, and why is it better than traditional approaches such as probabilistic models are still largely unknown. In this paper, we focus on the context of topic modeling and highlight a key advantage of self-supervised learning - when applied to data generated by topic models, self-supervised learning can be oblivious to the specific model, and hence is less susceptible to model misspecification. In particular, we prove that commonly used self-supervised objectives based on reconstruction or contrastive samples can both recover useful posterior information for general topic models. Empirically, we show that the same objectives can perform on par with posterior inference using the correct model, while outperforming posterior inference using misspecified models.

## 1 Introduction

Recently researchers have successfully trained large-scale models like BERT (Devlin et al., 2018) and GPT (Radford et al., 2018), which offers extremely powerful representations for many NLP tasks (see e.g., Liu et al. (2021); Jaiswal et al. (2021) and references therein). To train these models, often one starts with sentences in a large text corpus, mark random words as "unknown" and ask the neural network to predict the unknown words. This approach is known as self-supervised learning (SSL).

Why can self-supervised approaches learn useful representations? To understand this we first need to define what are "useful representations". A recent line of work (Tosh et al., 2021a; Wei et al., 2021) studied self-supervised learning in the context of probabilistic models: assuming the data is generated by a probabilistic model (such as a topic model or Hidden Markov Model), one can define representation of observed data as the corresponding hidden variables in the model (such as topic proportions in topic models or hidden states in Hidden Markov Model). These works show that self-supervised learning approach is as good as explicitly doing inference using such models.

This approach naturally leads to the next question - why can self-supervised learning perform *better* than traditional inferencing based on probabilistic models? In this paper we study this question in the context of topic modeling, and highlight one key advantage for self-supervised learning: robustness to model misspecification.

Many different models (such as Latent Dirichlet Allocation (LDA) (Blei et al., 2003), Correlated Topic Model (CTM) (Blei & Lafferty, 2007), Pachinko Allocation Model (PAM) (Li & McCallum, 2006)) have been applied in practice. Traditional approaches would require different ways of doing inference *depending on* which model is used to generate the data. On the other hand, we show that no matter which topic model is used to generate the data, if standard self-supervised learning objectives such as the reconstruction-based objective[1] or the contrastive objective[2] can be minimized, then they will generate representations that contain useful information about the topic proportions of a document. Self-supervised learning is *oblivious* to the choice of the probabilistic model, while the traditional approach of probabilistic modeling depends highly on the specific model. Therefore, one

---

[*]Equal contribution.
[1]See Equation (1), similar to the objective used in Pathak et al. (2016); Devlin et al. (2018)
[2]See Equation (2), this was also used in Tosh et al. (2021a).

would expect self-supervised learning to perform similarly to inferencing with the correct model, and outperforms inferencing with misspecified model.

To verify our theory, we run synthetic experiments to show that self-supervised learning indeed outperforms inferencing with misspecified models. Unlike large-scale models, our self-supervised learning is applied in the much simpler context of topic models, but we also demonstrate that even this simple application can improve over simple baselines on real data.

## 1.1 RELATED WORKS

**Self-Supervised Learning** Self-supervised learning recently has been shown to be able to learn useful representation, which is later used for downstream tasks. See for example Bachman et al. (2019); Caron et al. (2020); Chen et al. (2020a;b;c); Grill et al. (2020); Chen & He (2021); Tian et al. (2020a); He et al. (2020) and references therein. In particular, Devlin et al. (2018) proposed BERT, which shows that self-supervised learning has the ability to train large-scale language models and could provide powerful representations for downstream natural language processing tasks.

**Theoretical Understanding of Self-Supervised Learning** Given the recent success of self-supervise learning, many works have been tried to provide theoretical understanding on contrastive learning (Arora et al., 2019; Wang & Isola, 2020; Tosh et al., 2021a; Tian et al., 2020b; HaoChen et al., 2021; Wen & Li, 2021; Zimmermann et al., 2021) and reconstruction-based learning (Lee et al., 2020; Saunshi et al., 2020; Teng & Huang, 2021). Also, several papers considered the problem from a multi-view perspective (Tsai et al., 2020; Tosh et al., 2021b), which covers both contrastive and reconstruction-based learning. Moreover, Wei et al. (2020) and Tian et al. (2021) studied the theoretical properties of self-training and the contrastive learning without the negative pairs respectively. Saunshi et al. (2020) investigated the benefits of pre-trained language models for downstream tasks. Most relevant to our paper, Tosh et al. (2021a) considered the contrastive learning in the topic models setting. Our theoretical results extend their theory to reconstruction-based objective (while also removing some assumptions for the contrastive objective), and our empirical results show that the reconstruction-based objective can be effectively minimized.

**Theoretical Analysis of Topic Models** Many works have proposed provable algorithms to learn topic models, such as methods of moment based approaches (Anandkumar et al., 2012; 2013; 2014; 2015) and anchor word based approaches (Papadimitriou et al., 2000; Arora et al., 2012; 2016a; Gillis & Vavasis, 2013; Bittorf et al., 2012). Much less is known about provable inference for topic models. Sontag & Roy (2011) showed that MAP estimation can be NP-hard even for LDA model. Arora et al. (2016b) considered approximate inference algorithms.

## 1.2 OUTLINE

We first introduce the basic concepts of topic models and our objectives in Section 2. Then in Section 3 we prove guarantees for the reconstruction-based objective. Section 4 connects the contrastive objective to reconstruction-based objective which allows us to prove a stronger guarantee for the former. We then demonstrate the ability of self-supervised learning to adapt to different models by synthetic experiments in Section 5. Finally, we also evaluate the reconstruction-based objective on real-data to show that despite the simplicity of the topic modeling context, it extracts reasonable representations in Section 6.

## 2 PRELIMINARIES

In this section we first introduce some general notations. Then we briefly describe the topic models we consider in Section 2.1. Finally we define the self-supervised learning objectives in Section 2.3 and give our main results.

**Notation** We use $[n]$ to denote set $\{1, 2, \ldots, n\}$. For vector $x \in \mathbb{R}^d$, denote $\|x\|$ as its $\ell_2$ norm and $\|x\|_1$ as its $\ell_1$ norm. For matrix $A \in \mathbb{R}^{m \times n}$, we use $A_i \in \mathbb{R}^m$ to denote its $i$-th column. When matrix $A$ has full column rank, denote its left pseudo-inverse as $A^\dagger = (A^\top A)^{-1} A^\top$. For matrix or general tensor $T \in \mathbb{R}^{d_1 \times \cdots \times d_l}$, we use vector $\text{vec}(T) \in \mathbb{R}^{d_1 \cdots d_l}$ to represent its vectorization. Let $\mathcal{S}_k = \{x \in \mathbb{R}^k | \sum_{i=1}^k x_i = 1, x_i \geq 0\}$ to denote $k - 1$ dimensional probability simplex. For two probability vectors $p, q$, define their total variation (TV) distance as $\text{TV}(p, q) = \|p - q\|_1 / 2$.

## 2.1 Topic Models

Many topic models treat documents as a bag-of-words and use a two-step procedure of generating a document: first each document is viewed as a probability distribution of topics (often called the topic proportions $w$), and each topic is viewed as a probability distribution over words. To generate a word, one first selects a topic based on the topic proportions $w$, and then samples a word from that particular topic. Since the topic distributions are shared across the entire corpus, and the topic proportion vector $w$ is specific to documents, it makes sense to define $w$ as the representation for the document.

More precisely, let $\mathcal{V}$ be a finite vocabulary with size $V$ and $\mathcal{K}$ be a set of $K$ topics, where each topic is a distribution over $\mathcal{V}$. We denote the topic-word matrix as $A$ with $A_{ij} = \mathbb{P}(\text{word } i \mid \text{topic } j)$, so that each column in $A$ represents the word distribution of a topic. Each document corresponds to a convex combination of different topics with proportions $w$. For each word in the document, first sample a topic $z$ according to the topic proportions $w$, and then sample the word from the corresponding topic (equivalently, one can also sample a word from distribution $Aw$). Different topic models differ in how they generate $w$, which we formulate in the following definition:

**Definition 2.1** (General Topic Model). A general topic model specifies a distribution $\Delta(K)$ for each number of topics $K$. Given a topic-word matrix $A$ and $\Delta(K)$, to generate a document, one first sample $w \sim \Delta(K)$ and then sample each word in the document from the distribution $Aw$.

Here $\Delta(K)$ is a prior distribution of $w$ which is crucial when trying to infer the true topic proportions $w$ given the words in a document. Of course, different topic models may also specify different priors for the topic-word matrix $A$. However, given a large number of documents generated from the topic model, in many settings one can hope to learn the topic-word matrix $A$ and the prior distribution $\Delta(K)$ (see e.g., Arora et al. (2012; 2016a)), therefore we consider the following inference problem:

**Definition 2.2** (Topic Inference). Given a topic-word matrix $A$, prior distribution $\Delta(K)$ for topic proportions $w$, and a document $x$, the topic inference problem tries to compute the posterior distribution $w|(x, A)$ given the prior $w \sim \Delta(K)$.

Note that our general topic model can capture many standard topic models, including pure topic model, LDA, CTM and PAM.

## 2.2 Semi-supervised Learning Setup

Our goal is to understand the representation learning aspect of self-supervised learning, where we are given large number of unlabeled documents and a small number of labeled documents. In the unsupervised learning stage, the algorithm only has access to the unlabeled documents, and the goal is to learn a representation (a mapping from documents to a feature vector). In the supervised learning stage, the algorithm would use the labeled documents to train a simple classifier that maps the feature vector to the actual label. We often refer to the unsupervised learning stage as "feature learning" and the supervised learning stage as "downstream application".

In the theoretical model, we assume that the label $y$ is a simple function of the topic proportions $w$. The goal is to apply self-supervised learning on the unlabeled documents to get a good representation, and then use this representation to predict the label $y$. Of course, if one knows the actual parameters of the model, the best predictor for $y$ would be to first estimate the posterior distribution $w$ and then apply the function that maps $w$ to label $y$. We will show that for functions that are approximable by low degree polynomials self-supervised learning can always provide a good representation.

## 2.3 Self-supervised learning

In general, self-supervised learning tries to turn unlabeled data into a supervised learning problem by hiding some known information. There are many ways to achieve this goal (see e.g., Liu et al. (2021); Jaiswal et al. (2021)). In this paper, we focus on two different approaches for self-supervised learning: reconstruction-based objective and contrastive objective. We first formally define the objectives and then discuss our corresponding result.

**Reconstruction-Based Objective** One common approach of self-supervised learning is to first mask part of the data and then try to find a function $f$ to reconstruct the missing part given the unmasked part of input. This is commonly used for language modeling (Devlin et al., 2018; Radford et al.,

2018). In the context of topic modelling, suppose all the documents are generated from an underlying topic model with $A$ and $\Delta(K)$. Then for any given document $x_{\text{unsup}}$, since each word is i.i.d. sampled (so that the order of words is not important), we pick $t$ random words from the document and mark them as unknown, then we ask the learner to predict these $t$ words given the remaining words in the document. Specifically, we split $x_{\text{unsup}}$ into $x$ and $y$ where $y$ is the $t$ words that we select and let $x$ is the document with these $t$ words removed. We aim to select a predictor $f$ that minimizes the following reconstruction objective:

$$\min_f L_{\text{reconst}}(f) \triangleq \mathbb{E}_{x,y}[\ell(f(x), y)], \tag{1}$$

where $\ell(\hat{y}, y) = \sum_k -y_k \log \hat{y}_k$ is the cross entropy loss. Here we slightly abuse the notation to use $y$ as an one-hot vector for $V^t$ classes and $f$ also outputs a probability vector on $\mathcal{V}^t$. Depending on the context, we will use $y$ to denote either the actual next $t$ words or its corresponding one-hot label. Now we are ready to present our result for the reconstruction-based objective:

**Theorem 2.3.** *(Informal) Consider the general topic model setting as Definition 2.1, suppose function $f$ minimizes the reconstruction-based objective* (1)*. Then, any polynomial $P(w)$ of the posterior $w|(x, A)$ with degree at most $t$ can be represented by a linear function of $f(x)$.*

The theorem shows that if we want to get basic information about posterior distribution (such as mean, variance), then it suffices to predict a small constant number of words (1 for mean and 2 for variance). See Theorem 3.1 for the formal statement.

**Contrastive Objective** Another common approach in self-supervised learning is the contrastive learning (He et al., 2020; Chen et al., 2020a). In contrastive learning, the training data is usually a pair of data $(x, x')$ with a label $y \in \{0, 1\}$, where label 1 means $(x, x')$ is a positive sample ($x$ and $x'$ are similar) and label 0 means $(x, x')$ is a negative sample ($x$ and $x'$ are dissimilar). The task is to find a function $f$ such that it can distinguish the positive sample and negative sample, i.e., $f(x, x') = y$. Formally, we want to select a predictor $f$ such that it minimizes the following contrastive objective:

$$\min_f L_{\text{contrast}} \triangleq \mathbb{E}_{x,x',y}[\ell(f(x, x'), y)]. \tag{2}$$

In the context of topic models, following previous work (Tosh et al., 2021a), we generate the data $(x, x', y)$ as follows. Suppose all documents are generated from an underlying topic model with $A$ and $\Delta(K)$. We first generate a document $x$ from the word distribution $Aw$ where $w$ is sampled from $\Delta(K)$. Then, (i) with half probability we generate $t$ words from the same distribution $Aw$ to form the document $x'$ and set $y = 1$; (ii) with half probability we generate $t$ words from a different word distribution $Aw'$ with $w' \sim \Delta(K)$ (so that $w \neq w'$) and set $y = 0$. We consider the square loss $\ell(\hat{y}, y) = (\hat{y} - y)^2$ for simplicity.

We now give our informal result on contrastive objective. See Theorem 4.1 for the formal statement. Note that this theorem generalizes Theorem 3 in Tosh et al. (2021a).

**Theorem 2.4.** *(Informal) Consider the general topic model setting as Definition 2.1, suppose function $f$ minimizes the contrastive objective* (2)*. Then we can use $f$ and enough documents to construct a representation $g(x)$ such that any polynomial $P(w)$ of the posterior $w|(x, A)$ with degree at most $t$ can be represented by a linear function of $g(x)$.*

**Advantage of SSL approach** Note that neither objectives (1) or (2) depend on the prior distribution $\Delta(K)$, so it is possible to optimize these without specifying a specific model; on the other hand, even the definition of topic inference (Definition 2.2) relies heavily on $\Delta(K)$.

## 3 GUARANTEES FOR THE RECONSTRUCTION-BASED OBJECTIVE

In this section, we consider the reconstruction-based objective (1) and provide theoretical guarantees for its performance. We first show that if such objective with $t$ unknown words can be minimized, then any polynomial of topic posterior $w|(x, A)$ with degree at most $t$ can be represented by a linear function of the learned representation.

**Theorem 3.1** (Main Result)**.** *Consider the general topic model setting as Definition 2.1, suppose topic-word matrix $A$ satisfies $\text{rank}(A) = K$ and function $f$ minimizes the reconstruction-based objective* (1)*. Then, for any document $x$ and its posterior $w|(x, A)$, the posterior mean of any*

*polynomial $P(w)$ with degree at most $t$ is linear in $f(x)$. That is, there exists a $\theta \in \mathbb{R}^{V^t}$ such that for all documents $x$*

$$\mathbb{E}_w[P(w)|x, A] = \theta^\top f(x).$$

To understand this theorem, first consider a warm-up example where $t = 1$ (see details in Section A.1 in appendix). Intuitively, in this case the best way to predict the missing word for a given document $x$ is to estimate its topic proportion $w$, and then predict the word using $Aw$ where $A$ is the topic-word matrix. Therefore, if the output of self-supervised learning $f(x)$ (which is a $V$-dimensional vector indexed by words) minimizes the loss, then $f(x)$ must have the form $f(x) = A\mathbb{E}[w|x, A]$. When $A$ is full rank multiplying by the pseudo-inverse of $A$ recovers the expectation of the posterior $w|(x, A)$. The proof for the general case of predicting $t$-words requires more careful characterization of the optimal prediction and its relationship to $\mathbb{E}_w[P(w)|x, A]$, which we defer to Section A.2.

The above discussion also suggests that though the representation dimension may seem to be $V^t$, its "effective" dimension is in fact $K^t$ which is much smaller than $V^t$. See more details in Appendix A.4.

**Robustness for an approximate minimizer**     In Theorem 3.1 we focus on the case when function $f$ is exactly the minimizer of the reconstruction-based objective (1), i.e., $L_{\text{reconst}}(f) = L^*_{\text{reconst}} \triangleq \min_f L_{\text{reconst}}(f)$. However, in practice one cannot hope to find such a function exactly. In the following, we provide a robust version of Theorem 3.1 such that it allows us to find an approximate solution instead of the exact optimal solution.

To present our result, we need to first introduce the following notion of condition number, which was used in many previous works, such as collaborative filtering systems (Kleinberg & Sandler, 2008) and topic models (Arora et al., 2016b). Intuitively, $\kappa(B)$ measures how large a vector would change after multiplying with $B$ in the $\ell_1$ norm sense.

**Definition 3.2** ($\ell_1$ Condition Number). *For matrix $B \in \mathbb{R}^{m \times n}$, define its $\ell_1$ condition number $\kappa(B)$ as $\kappa(B) \triangleq \min_{x \in \mathbb{R}^n : x \neq 0} \|Bx\|_1 / \|x\|_1 = \max_{i \in [n]} \|B_i\|_1$, where $B_i$ is the $i$-th column of $B$.*

Let $W_{post} \in \mathbb{R}^{K \times \cdots \times K}$ be the topic posterior tensor for $t$ unknown words $y = (y_1, \ldots, y_t)$ given remaining document $x$. That is, for each entry $[W_{post}]_{z_1, \ldots, z_k} = \mathbb{P}(\text{for all } i, z_i \text{ is the topic of word } y_i|x, A) = \mathbb{E}_w[w_{z_1} \cdots w_{z_t}|x, A]$ and $W_{post} = \mathbb{E}_w[w^{\otimes t}|x, A]$.[3] Thus, any polynomial $P(w)$ of degree at most $t$ can be represented as $\mathbb{E}_w[P(w)|x, A] = \beta^\top \text{vec}(W_{post})$ for some $\beta$, where $\text{vec}(W_{post}) \in \mathbb{R}^{K^t}$ is the vectorization of $W_{post}$.

We now are ready to present the robust version of Theorem 3.1. It shows if we only find a function $f$ whose loss is at most $\epsilon$ larger than the optimal loss, then a linear transformation of our learned representation can still give a good approximation of the target polynomial within a $O(\epsilon)$ error.

**Theorem 3.3** (Robust Version). *Consider the general topic model setting as Definition 2.1, suppose topic-word matrix $A$ satisfies $rank(A) = K$ and function $f$ satisfies $L_{reconst}(f) \leq L^*_{reconst} + \epsilon$ for some $\epsilon > 0$. Then, for any document $x$ and its posterior $w|(x, A)$, the posterior mean of any polynomial $P(w)$ with degree at most $t$ is approximately linear in $f(x)$. That is, there exists a $\theta \in \mathbb{R}^{V^t}$ such that*

$$\mathbb{E}_x\left[\left(\mathbb{E}_w[P(w)|x, A] - \theta^\top f(x)\right)^2\right] \leq 2 \|\beta\|^2 \kappa^{2t}(A^\dagger)\epsilon,$$

*where $\mathbb{E}_w[P(w)|x, A] = \beta^\top vec(W_{post})$.*

Note that the dependency on $\|\beta\|^2 \kappa^{2t}(A^\dagger)$ is expected, since this is the norm $\|\theta\|^2$ that we would have if $\epsilon = 0$, i.e., the $\theta$ we would have in Theorem 3.1. Thus, this quantity should be understood as the complexity of the target function for the downstream task. Empirically we show that $\kappa(A^\dagger)$ is small in Section C.6. The proof is deferred to Section A.3.

## 4    GUARANTEES FOR THE CONTRASTIVE OBJECTIVE

In this section, we consider the contrastive objective (2) for self-supervised learning and provide similar provable guarantees on its performance as the reconstruction-based objective.

---

[3]To simply the notations, WLOG assume topic set $\mathcal{K} = [K]$ so that we can use topics $z_i \in [K]$ as indices.

We use the same approach as Tosh et al. (2021a) to construct a representation $g(x)$. Given a set of landmark documents $\{l_i\}_{i=1}^m$ with length $|l_i| = t$ as references, the representation is defined as

$$g(x, \{l_i\}_{i=1}^m) = (g(x, l_1), \ldots, g(x, l_m))^\top, \quad g(x, x') = \frac{f(x, x')}{1 - f(x, x')}. \tag{3}$$

The following theorem gives the theoretical guarantee for this representation. Similar to Theorem 3.1 for reconstruction-based objective, it shows that any polynomial $P(w)$ of degree at most $t$ can be represented by a linear function of the learned representation. Our proof here relies on the observation that having $g(x, \{l_i\}_{i=1}^m)$ for all short landmark documents of length $t$ gives similar information as $f$ in the reconstruction-based objective does. This observation allows us to remove the anchor words assumption needed in (Tosh et al., 2021a).

**Theorem 4.1.** *Consider the general topic model setting as Definition 2.1, suppose topic-word matrix $A$ satisfies $rank(A) = K$ and function $f$ minimizes the contrastive objective* (2). *Assume we randomly sampled $m = K^t$ different landmark documents $\{l_i\}_{i=1}^m$ and construct $g(x, \{l_i\}_{i=1}^m)$ as* (3). *Then, for any document $x$ and its posterior $w|(x, A)$, the posterior mean of any polynomial $P(w)$ with degree at most $t$ is linear in $g(x, \{l_i\}_{i=1}^m)$. That is, there exists $\theta \in \mathbb{R}^m$ such that for all documents $x$*

$$\mathbb{E}_w[P(w)|x, A] = \theta^\top g(x, \{l_i\}_{i=1}^m).$$

In fact, one can use the same set of documents for both representation and downstream task so that we do not need additional landmark documents.

## 5 SYNTHETIC EXPERIMENTS

In this section, we optimize the reconstruction-based objective (1) on data generated by several topic models to show that self-supervised learning performs on par with inference using correct model, and both of them outperform inference with misspecified models.

### 5.1 TOPIC MODELS

We consider four types of topic models in our experiments. Our first topic model is the pure-topic model, where each document's topic comes from a discrete uniform distribution over the $K$-dimensional linear basis, namely $\{e_1, e_2, ..., e_K\}$. Our second topic model is the Latent Dirichlet Allocation (LDA) model, where $\Delta(K)$ is a symmetric Dirichlet distribution $\text{Dir}(1/K)$.

We are also interested in topic models that involve more subtle topic correlations and similarities between different topic's word distributions. To this end, we consider the Correlated Topic Model (CTM) (Blei & Lafferty, 2007) and the Pachinko Allocation Model (PAM) (Li & McCallum, 2006). Our goal is to construct settings where the correlation between topics provide useful information for inference. To achieve this goal, in CTM, we construct groups of 4 topics. Within the group, topic pairs (0,2) and (1,3) are highly correlated (as specified by the prior), while topics 0 and 1 share many words (see Figure 1 for an illustration of the setting). In this case, if we observe a document with words that could either belong to topic 0 or 1, but this same document also has words that are associated with topic 2, we can infer that the first set of words are likely from topic 0, not topic 1. We construct such correlations in CTM by setting the diagonal entries of its Gaussian covariance matrix to 15 and the covariance between correlated pairs of topics to 0.99 times the diagonal entries, with the remaining entries set to zero. We construct similar examples for PAM, see Appendix C for details.

### 5.2 SIMULATION SETUP

**Document generation** Our documents are synthetically generated through the following steps: we first construct a $V \times K$ topic-word matrix $A$. Then, for each document, we determine the document length $n$ from Poisson distribution $\text{Pois}(\lambda)$, draw a topic distribution $w$ from $\Delta(K)$, and draw $n$ words i.i.d. from the word distribution given by $Aw$. In our simulation, we set $K = 20, V = 5000, \lambda \in \{30, 60\}$. Often $A$ will be drawn from a Dirichlet distribution $\text{Dir}(\alpha/K)$ (see Appendix C for details), and we take $\alpha = 1, 3, 5, 7, 9$ to vary the difficulty level of the inference problem, where a larger $\alpha$ introduces more similarity between topics.

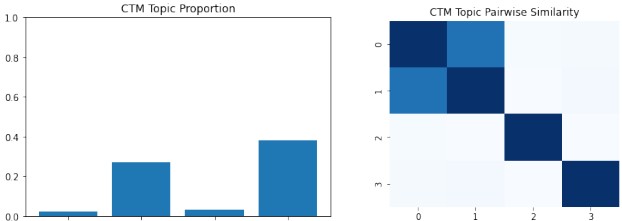

Figure 1: One example of a group of 4 topics in the Correlated Topic Model. **Left:** Weight of each topic in a document's topic proportion. In this example topics 1 and 3 have large proportions as they are correlated. **Right:** Pairwise topic similarity (cosine similarity of the pair's word distribution).

**Neural network models**    We consider two different network architectures as $f$ in the experiments: (1) Fully-connected network with residual connection, with bag-of-words document representation as the input since the order does not matter in topic models; (2) Attention-based architecture. Due to the architecture of attention, the input needs to be a sequence of words instead of bag-of-words vector. Thus, we transform bag-of-words representation into full document by repeating each word by its frequency and concatenating them in random order, and then use the full document as the input. More specifically, the attention-based architecture contains 8 transformer blocks where each transformer block consists of a 768-dimensional attention layer and a feed-forward layer, with residual connection applied around every block. The network's second to last layer averages over the outputs of the last transformer block and the final layer projects it to a $V$-dimensional word distribution. See more details in Section C.5. We find that fully-connected neural network with residual connections performs the best on recovering topic posterior distribution for pure-topic and LDA documents, and attention-based architecture (Vaswani et al., 2017) performs the best for recovering topic posterior distribution for CTM and PAM documents.

**Training setup**    During training, we resample 60,000 new documents after every 2 epochs, and the total amount of training data varies from 720K documents to 6M documents. The loss function is the reconstruction-based objective (1) with $t = 1$, i.e., we want the model to predict one missing word. To reduce the variance during the training, in each training document, we sampled 6 words as the prediction target, hide them from the model and ask the model to predict each word separately. The trained model is evaluated on test documents generated from the same topic prior as the training documents. We use 5,000 test documents for the pure-topic prior and 200 test documents for the remaining priors. For each test document, we use MCMC (see Appendix C.9) assuming the document's correct topic prior to approximate the ground truth topic posterior. To get the posterior estimations from SSL approach, we multiply the pseudo-inverse of the topic matrix $A$ with the model output $f(x)$ to get the estimated posterior mean vector $A^\dagger f(x)$ as we explained in Section 3 (and also Section A.1). Then, we measure topic posterior recovery loss as the Total Variation (TV) distance between the recovered topic posterior mean vector and the ground truth topic posterior mean vector.

### 5.3    Experiment Results

**Topic posterior recovery**    As illustrated in Table 1, after 200 training epochs, our model can accurately recover the topic posterior mean vector. Meanwhile, it can be observed that for larger values of the Dirichlet hyperparameter $\alpha$, the recovery loss gets higher. This is expected because higher $\alpha$ leads to more similar topics, which makes the learning problem more difficult. This effect is also captured by Definition 3.2 and we show the computed condition numbers in Section C.6.

**Major topic recovery**    We examine the extent our recovered topic posterior captures the major topics in a given document. For pure-topic and LDA documents, we measure major topic recovery accuracy of correctly estimating the topic with largest proportion. Considering that topics in CTM and PAM documents are correlated in pairs, we measure the major topic recovery accuracy for CTM and PAM as the top-2 topic overlap rate on their test documents. Table 1 shows that our algorithm is successful throughout all settings we consider.

**Robustness of self-supervised learning**    We compare the self-supervised approach to traditional topic inference (MCMC with a specifed topic prior, the results on variational inference are reported

| TV Distance $\alpha$ | Document Type | | | |
|---|---|---|---|---|
| | Pure | LDA | CTM | PAM |
| 1 | 0.0148 ± 0.0020 | 0.0757 ± 0.0033 | 0.0550 ± 0.0037 | 0.0489 ± 0.0025 |
| 3 | 0.0308 ± 0.0076 | 0.0899 ± 0.0053 | 0.0799 ± 0.0048 | 0.0712 ± 0.0038 |
| 5 | 0.0501 ± 0.0030 | 0.1041 ± 0.0062 | 0.0970 ± 0.0057 | 0.0787 ± 0.0043 |
| 7 | 0.0391 ± 0.0022 | 0.1233 ± 0.0071 | 0.1071 ± 0.0068 | 0.0960 ± 0.0057 |
| 9 | 0.0517 ± 0.0045 | 0.1358 ± 0.0089 | 0.1101 ± 0.0064 | 0.0971 ± 0.0053 |

| Major Topic(s) Recovery Accuracy $\alpha$ | Document Type | | | |
|---|---|---|---|---|
| | Pure | LDA | CTM | PAM |
| 1 | 1.0000 ± 0.0000 | 0.9050 ± 0.0406 | 0.9175 ± 0.0275 | 0.9025 ± 0.0330 |
| 3 | 1.0000 ± 0.0000 | 0.8750 ± 0.0458 | 0.8900 ± 0.0311 | 0.8950 ± 0.0344 |
| 5 | 1.0000 ± 0.0000 | 0.9000 ± 0.0416 | 0.8975 ± 0.0296 | 0.8675 ± 0.0407 |
| 7 | 1.0000 ± 0.0000 | 0.9150 ± 0.0387 | 0.8675 ± 0.0383 | 0.8675 ± 0.0407 |
| 9 | 0.9950 ± 0.0098 | 0.9100 ± 0.0397 | 0.8850 ± 0.0330 | 0.8325 ± 0.0461 |

Table 1: Self-supervised learning approach performs reasonably on recovering the topic posterior mean. **Left:** TV distance between recovered topic posterior (self-supervised learning approach) and true topic posterior for different topic models. **Right:** Major topic(s) recovery accuracy for different topic models. The 95% confidence interval is reported in both tables.

| TV Distance Method | Document Type ($\alpha = 1$) | | | |
|---|---|---|---|---|
| | Pure | LDA | CTM | PAM |
| LDA | 0.0406 ± 0.0016 | - | 0.1182 ± 0.0099 | 0.1218 ± 0.0096 |
| CTM | 0.2083 ± 0.0038 | 0.2060 ± 0.0064 | - | 0.3154 ± 0.0082 |
| PAM | 0.3782 ± 0.0038 | 0.3459 ± 0.0128 | 0.3939 ± 0.0096 | - |
| **SSL (ours)** | **0.0148 ± 0.0020** | **0.0757 ± 0.0033** | **0.0550 ± 0.0037** | **0.0489 ± 0.0025** |

| Major Topic(s) Recovery Method | Document Type ($\alpha = 1$) | | |
|---|---|---|---|
| | LDA | CTM | PAM |
| LDA | **0.9150 ± 0.0387** | 0.8850 ± 0.0344 | 0.8500 ± 0.0360 |
| CTM | 0.9000 ± 0.0416 | **0.9175 ± 0.0275** | 0.8300 ± 0.0370 |
| PAM | 0.8750 ± 0.0458 | 0.7250 ± 0.0397 | **0.9050 ± 0.0320** |
| **SSL (ours)** | 0.9050 ± 0.0406 | 0.9175 ± 0.0275 | 0.9025 ± 0.0330 |

Table 2: Robustness of self-supervised learning approach. **Left:** TV distance between recovered topic posterior and true topic posterior of self-supervised learning approach versus posterior inference via Markov Chain Monte Carlo assuming a specific prior for $\alpha = 1$. **Right:** Major topic recovery rate of our approach versus posterior inference via Markov Chain Monte Carlo assuming a specific prior for $\alpha = 1$. In both table, the 95% confidence interval is reported.

.

in Section C.3). Specifically, for each $\alpha$ value, we take 200 test documents from each category of documents, and we run posterior inference assuming a specific topic model and calculate the TV distance between this posterior mean vector and the ground truth posterior mean vector. We exclude assuming pure-topic prior from our comparison because it gives invalid results for documents with mixed topics. For $\alpha = 1$, as shown in Table 2, the topic posterior recovered from SSL approach is closer to the ground truth topic posterior than that recovered from a misspecified topic prior.

We also report the major topic recovery accuracy for SSL approach against topic inference using both correct and incorrect topic model in Table 2. The major topic recovery rate of SSL approach is similar to that of posterior inference assuming the correct prior. For pure topic model, all four methods get 100% major topics recovery rate on pure-topic documents. For LDA, again all four methods perform similarly well since the instance is not difficult. However we observe significant difference for more complicated CTM and PAM models. For these models, the SSL approach performs similarly to posterior inference using the correct model, and both of them are significantly better than posterior inference using misspecified models. The results for $\alpha = 3, 5, 7, 9$ are presented in Section C. The comparison further reveals the robustness of self-superivised learning.

# 6 EXPERIMENTS ON REAL DATA

The self-supervised learning approach studied in this paper is simplified for theoretical analysis as it uses a bag-of-words approach. In this section, we show that although this simplified approach is not state-of-art, it has reasonable performance compared to several baselines.

**Experiment Setup** We use the AG news dataset (Zhang et al., 2015), in which each document has a label of one out of four categories: world, sports, business, and sci/tech. Each category has 30,000 samples in the training set and 19,000 samples in the testing set. The number of words in the vocabulary is about 16,700. We generally follow the experiment setup done by Tosh et al. (2021a) (see more details in Section D.1), but the representation we used is generated by our reconstruction-based objective. Experiment results on two additional datasets are provided in Appendix D.

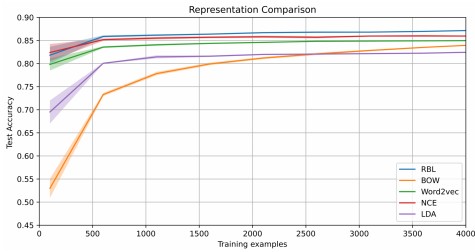

Figure 2: Performances of RBL, NCE and baselines (BOW, Word2vec, LDA) on real data.

Our training follows the semi-supervised setup that involves two major phases: unsupervised phase and supervised phase. In the unsupervised phase, we train the self-supervised model with reconstruction-based objective on most of training data. In the supervised phase, using the document representation generated by the SSL model as input, we train a linear classifier to classify the category of the documents using multi-class logistics regression (by Scikit-learn (Pedregosa et al., 2011)) with 3-fold cross validation for parameter tuning ($l_2$ regularization term and solver). We measure the topic recovery accuracy on test set to compare the performance of different methods.

We choose residual blocks as the basic building block for our neural network architecture, with varying width and depth. The network used for Figure 2 has 3 residual blocks and a width of 4096 neurons per layer. Other detailed hyperparameters and ablation studies can be found in Appendix D.3. We train the model using reconstruction-based objective (1) with $t = 1$ using the same variance-reduction technique in synthetic experiments.

To extract our representation, we use an identity matrix to replace the last layer. That is, we take a softmax function on top of the second-to-last layer of the neural network. This is different from our theory but we found that it effectively reduces the high output dimension and improves performance. We include more details in Section D.2 in Appendix.

**Experiment Results** The performance of Reconstruction-Based Learning representation (RBL) is shown in Figure 2, where the test accuracy on representation is plotted against number of training samples used to train the classifier in supervised learning phase. The representation RBL performs better than both bag-of-words (BOW),Word2vec and LDA baselines (see more details about baselines in Section D.2). Notably, Word2vec representation is inferior only by a small margin, and works well in general even when limited training samples are provided. On the contrary, although BOW representation has a decent test accuracy when training samples are abundant, it performs significantly poorly on smaller training set.

We also compare our result to previous noise contrastive estimation (NCE) representation benchmark achieved by Tosh et al. (2021a) using contrastive objective, where their reported best accuracy is around $87.5\%$ when full 4000 training samples are used, slightly higher than our RBL corresponding test accuracy of $87.1\%$. In our attempt to reproduce their result, parameter tuning yields the best accuracy of $86\%$ when full training samples were used. We plot our own reproduced results on in Figure 2 since parameter tuning does not achieve a close benchmark to their result.

## 7    CONCLUSION AND LIMITATIONS

"All models are wrong but some are useful." If one self-supervised objective can capture all models, then it would be able to extract useful information. In this paper, we studied the self-supervised learning in the topic models setup and showed that it can provide useful information about the topic posterior no matter what topic model is used. Our results generalized previous work (Tosh et al., 2021a) to both contrastive learning and reconstruction-based learning and our techniques allow us to depend on weaker assumptions. We also empirically showed that the reconstruction-based learning performs better than the posterior inference under misspecified models, and it can provide useful representation for the topic inference problem. Our theoretical analysis is limited to a bag-of-words setting, which also limits its empirical performance. Extending the work to more complicated models is an immediate open problem.

ACKNOWLEDGEMENT

This work is supported by NSF Award DMS-2031849, CCF-1845171 (CAREER), CCF-1934964 (Tripods) and a Sloan Research Fellowship.

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

## A  OMITTED PROOFS IN SECTION 3

In this section, we give the omitted proofs in Section 3. We first give a warmup example to illustrate our proof idea in Section A.1. Then we give the proof of our main result (Theorem 3.1) in Section A.2 and the proof of robust version (Theorem 3.3) in Section A.3.

### A.1  WARM-UP EXAMPLE: RECONSTRUCT THE UNKNOWN WORD

In this warm-up example, we consider the simple setting where we try to predict the only one unknown word of the document using reconstruction-based objective (1). The following result is the special case of Theorem 3.1 with $t = 1$, which shows that the learned representation is able to give any linear function of the topic posterior.

**Theorem A.1.** *Consider the general topic model setting as Definition 2.1, suppose topic-word matrix $A$ satisfies rank$(A) = K$ and function $f$ minimizes the reconstruction-based objective (1). Then, for any document $x$ and its posterior $w|(x, A)$, any linear function $P(w)$ is linear in $f(x)$, that is for any $P(w) = \beta^\top w$, there exists a $\theta \in \mathbb{R}^V$ such that for all documents $x$*

$$\mathbb{E}_w \left[ \beta^\top w | x \right] = \theta^\top f(x).$$

Following the proof idea described in Section 3, we first give a characterization of the optimal function $f$. The following lemma shows that $f(x)$ must be the word posterior vector given the document $x$. The proof of this lemma is simply based on the property of the cross-entropy loss function and we defer it to Section A.1.1. Recall that our vocabulary set is $\mathcal{V} = \{v_1, \ldots, v_V\}$ and $x_{\text{unsup}} = (x, y)$ is the given document, where $x$ is unmasked part and $y$ is word marked as unknown.

**Lemma A.2.** *If $f$ minimizes the reconstruction-based objective (1), then we have for all document $x$*

$$f(x) = (\mathbb{P}(y = v_1 | x), \ldots, \mathbb{P}(y = v_V | x))^\top.$$

Based on the above lemma, we are able to prove Theorem A.1. It is easy to see that the word posterior distribution is $A\mathbb{E}_w[w|x]$, so we have $f(x) = A\mathbb{E}_w[w|x]$. Since the columns of $A$ are linearly independent, we know $\mathbb{E}_w[w|x] = A^\dagger f(x)$. Thus, for any linear function $\beta^\top w$, there exists $\theta = (A^\dagger)^\top \beta$ such that $\mathbb{E}_w[\beta^\top w|x] = \theta^\top f(x)$. The formal proof of the general case is given in Section A.2.

#### A.1.1  PROOF OF LEMMA A.2

Instead of focusing on the $t = 1$ case, we directly give the corresponding lemma of Lemma A.2 for general $t$. Recall that $x_{\text{unsup}} = (x, y)$ is the given document, $x$ is the unmasked part, $y = (y_1, \ldots, y_t)$ is the $t$ unknown words that we want to predict and $\mathcal{V} = \{v_1, \ldots, v_V\}$ is the set of vocabulary. It shows that the optimal $f$ is the $t$ words posterior distribution. Note that the words posterior distribution is linear in the topic posterior of $t$-th moment, which is useful for the later analysis.

**Lemma A.3** (Lemma A.2, General Case)**.** *If $f$ minimizes the reconstruction-based objective (1), then we have for all document $x$*

$$f(x) = (\mathbb{P}(y = (v_1, v_1, \ldots, v_1)|x), \mathbb{P}(y = (v_2, v_1, \ldots, v_1)|x), \ldots, \mathbb{P}(y = (v_V, v_V, \ldots, v_V)|x))^\top.$$

*Proof.* Denote the word posterior distribution as $p^*(x)$. We will show $f(x) = p^*(x)$. By the law of total expectation, we have

$$L_{\text{reconst}} = \mathbb{E}_{x,y}[\ell(f(x), y)] = \mathbb{E}_x[\mathbb{E}_{y|x}[\ell(f(x), y)|x]].$$

We know the probability of $y$ given $x$ is $p^*(x)$. Since $\ell$ is cross-entropy loss, we have

$$L_{\text{reconst}} \geq \mathbb{E}_x \left[ - \sum_{k \in [V^t]} [p^*(x)]_k \log[p^*(x)]_k \right],$$

where the equality is obtained when $f(x) = p^*(x)$. Thus, when $f$ minimizes the reconstruction-based objective (1), we have $f(x) = p^*(x)$. $\square$

A.2 PROOF OF THEOREM 3.1

In this section, we give the proof of Theorem 3.1. Theorem A.1 is covered by Theorem 3.1 as the special case of $t = 1$. The proof follows the same idea as for Theorem A.1. We first show that $f(x)$ is the $t$-words posterior (Lemma A.3). Then just as $t = 1$ case where we can recover topic posterior with $A^\dagger f(x)$, we can also recover topic posterior of $t$-th moment with $\mathcal{A}^\dagger f(x)$ with some matrix $\mathcal{A}$. The result follows by the observation that for a polynomial $P(w)$ with degree at most $t$, $\mathbb{E}_w[P(w)|x]$ is a linear function of the topic posterior of $t$-th moment.

**Theorem 3.1** (Main Result). *Consider the general topic model setting as Definition 2.1, suppose topic-word matrix $A$ satisfies rank$(A) = K$ and function $f$ minimizes the reconstruction-based objective* (1). *Then, for any document $x$ and its posterior $w|(x, A)$, the posterior mean of any polynomial $P(w)$ with degree at most $t$ is linear in $f(x)$. That is, there exists a $\theta \in \mathbb{R}^{V^t}$ such that for all documents $x$*

$$\mathbb{E}_w[P(w)|x, A] = \theta^\top f(x).$$

*Proof.* Since $f$ is the minimizer of reconstruction-based objective (1), by Lemma A.3 we know given an input document $x$,

$$f(x) = (\mathbb{P}(y = (v_1, v_1, \ldots, v_1)|x), \mathbb{P}(y = (v_2, v_1, \ldots, v_1)|x), \ldots, \mathbb{P}(y = (v_V, v_V, \ldots, v_V)|x))^\top.$$

We will show that $f(x)$ is linear in the topic posterior. In the following, we focus on $[f(x)]_{y_1,\ldots,y_t}$, which is word posterior probability of the $t$ unknown words being $y_1, \ldots, y_t \in \mathcal{V}$ given document $x$. Recall $\mathcal{S}_K$ be the $K - 1$ dimensional probability simplex. By the law of total probability, we have

$$
\begin{aligned}
[f(x)]_{y_1,y_2,\ldots,y_t} &= \mathbb{P}(y_1, y_2, \ldots, y_t|x) \\
&= \int_{w \in \mathcal{S}_K} \mathbb{P}(y_1, y_2, \ldots, y_t, w|x) \mathrm{d}w \\
&= \int_{w \in \mathcal{S}_K} \mathbb{P}(y_1, y_2, \ldots, y_t|w, x) \mathbb{P}(w|x) \mathrm{d}w \\
&= \int_{w \in \mathcal{S}_K} \sum_{z_1,\ldots,z_t \in [K]} \mathbb{P}(y_1, y_2, \ldots, y_t, z_1, z_2, \ldots z_t|w) \mathbb{P}(w|x) \mathrm{d}w \\
&= \int_{w \in \mathcal{S}_K} \sum_{z_1,\ldots,z_t \in [K]} \mathbb{P}(z_1, , \ldots, z_t|w) \mathbb{P}(y_1, \ldots, y_t|z_1, \ldots, z_t, w) \mathbb{P}(w|x) \mathrm{d}w.
\end{aligned}
$$

Note that $z_i$ is the topic of word $y_i$. Since we consider the general topic model as Definition 2.1, we know

$$\mathbb{P}(y_1, y_2, \ldots, y_t|z_1, z_2, \ldots, z_t, w) = \mathbb{P}(y_1, y_2, \ldots, y_t|z_1, z_2, \ldots, z_t) = \prod_{i=1}^{t} \mathbb{P}(y_i|z_i) = \prod_{i=1}^{t} A_{y_i,z_i},$$

where $A$ is the topic-word matrix. Hence,

$$
\begin{aligned}
[f(x)]_{y_1,y_2,\ldots,y_t} &= \sum_{z_1,z_2,\ldots z_t \in [K]} \prod_{i=1}^{t} A_{y_i,z_i} \int_{w \in \mathcal{S}_K} \mathbb{P}(z_1, z_2, \ldots, z_t|w) \mathbb{P}(w|x) \mathrm{d}w \\
&= \sum_{z_1,z_2,\ldots z_t \in [K]} \prod_{i=1}^{t} A_{y_i,z_i} \int_{w \in \mathcal{S}_K} \prod_{i=1}^{t} w_{z_i} \mathbb{P}(w|x) \mathrm{d}w \\
&= \sum_{z_1,z_2,\ldots z_t \in [K]} \prod_{i=1}^{t} A_{x_{m+i},z_i} \mathbb{E}_w \left[ \prod_{i=1}^{t} w_{z_i} \middle| x \right].
\end{aligned}
$$

Recall that the topic posterior tensor is $W_{post} = \mathbb{E}_w[w^{\otimes t}|x] \in \mathbb{R}^{K \times \ldots \times K}$, where each entry $[W_{post}]_{z_1,\ldots,z_k} = \mathbb{P}(z_i$ is the topic of word $y_i$ for $i \in [t]|x) = \mathbb{E}_w[w_{z_1} \ldots w_{z_t}|x]$. Therefore,

$$[f(x)]_{y_1,y_2,\ldots,y_t} = \sum_{z_1,z_2,\ldots z_t \in [K]} \prod_{i=1}^{t} A_{y_i,z_i}[W_{post}]_{z_1,z_2,\ldots,z_t},$$

$$f(x) = (\underbrace{A \otimes A \otimes \cdots \otimes A}_{t \text{ times}})\text{vec}(W_{post}),$$

where $\otimes$ is the Kronecker product, $\mathcal{A} = A \otimes A \otimes \cdots \otimes A \in \mathbb{R}^{V^t \times K^t}$ and $\text{vec}(W_{post}) \in \mathbb{R}^{K^t}$ is the vectorization of $W_{post}$. Note that $\mathcal{A}^\dagger = A^\dagger \otimes \cdots \otimes A^\dagger$, so we have $\text{vec}(W_{post}) = \mathcal{A}^\dagger f(x)$.

Since $P(w)$ is a polynomial of degree at most $t$, we know there exists $\beta$ such that $\mathbb{E}_w[P(w)|x] = \beta^\top \text{vec}(W_{post})$. Thus, let $\theta = (\mathcal{A}^\dagger)^\top \beta$, we have $\mathbb{E}_w[P(w)|x] = \beta^\top \text{vec}(W_{post}) = \theta^\top f(x)$. □

### A.3 PROOF OF THEOREM 3.3

Similar to Lemma A.2, we can show that the representation $f(x)$ is close to $t$ words posterior. Then, the key lemma to obtain Theorem 3.3 is the following result. It suggests that if function $f$ is $\epsilon$-close to the optimal function $f^*$ (word posterior distribution) under cross-entropy loss, then we can still recover the topic posterior with proper $B$ up to $O(\sqrt{\epsilon})$ error. For example, we will choose $B = A^\dagger$ when $t = 1$. See the proof of Theorem 3.3 for the choice of $B$ in the general case.

**Lemma A.4.** *For cross-entropy loss $\ell$ and any two probability vectors $p, p^*$, if $\ell(p, p^*) \leq \min_{p:\sum_i p_i=1, p \geq 0} \ell(p, p^*) + \epsilon$, then for any matrix $B$ we have $\|Bp - Bp^*\|_1 \leq \kappa(B)\sqrt{2\epsilon}$.*

*Proof.* Recall that $\ell$ is cross-entropy loss, we know

$$\ell(p, p^*) - \min_{p:\sum_i p_i=1, p \geq 0} \ell(p, p^*) = D_{KL}(p \,||\, p^*) \leq \epsilon.$$

By Pinsker's Inequality, we have that

$$\|p - p^*\|_1 \leq \sqrt{2D_{KL}(p^* \,||\, p)} \leq \sqrt{2\epsilon}.$$

Then, for any matrix $B$ we have

$$\|Bp - Bp^*\|_1 \leq \kappa(B)\|p - p^*\|_1 \leq \kappa(B)\sqrt{2\epsilon},$$

where we use the definition of $\ell_1$ condition number (Definition 3.2). □

We now are ready to give the proof of Theorem 3.3, which is based on Lemma A.4 and the proof of Theorem 3.1.

**Theorem 3.3** (Robust Version). *Consider the general topic model setting as Definition 2.1, suppose topic-word matrix $A$ satisfies $rank(A) = K$ and function $f$ satisfies $L_{reconst}(f) \leq L^*_{reconst} + \epsilon$ for some $\epsilon > 0$. Then, for any document $x$ and its posterior $w|(x, A)$, the posterior mean of any polynomial $P(w)$ with degree at most $t$ is approximately linear in $f(x)$. That is, there exists a $\theta \in \mathbb{R}^{V^t}$ such that*

$$\mathbb{E}_x\left[\left(\mathbb{E}_w[P(w)|x, A] - \theta^\top f(x)\right)^2\right] \leq 2\|\beta\|^2 \kappa^{2t}(A^\dagger)\epsilon,$$

*where $\mathbb{E}_w[P(w)|x, A] = \beta^\top vec(W_{post})$.*

*Proof.* Recall from the proof of Theorem 3.1 that $\mathbb{E}_w[P(w)|x] = \beta^\top \text{vec}(W_{post}) = \beta^\top \mathcal{A}^\dagger f^*(x)$, where $\mathcal{A}^\dagger = A^\dagger \otimes \cdots \otimes A^\dagger$ and $\kappa(\mathcal{A}^\dagger) = \kappa^t(A^\dagger)$. Same as in the proof of Theorem 3.1, let $\theta = (\mathcal{A}^\dagger)^\top \beta$. We have

$$\mathbb{E}_x\left[\left(\mathbb{E}_w[P(w)|x] - \theta^\top f(x)\right)^2\right] = \mathbb{E}_x\left[\left(\beta^\top \mathcal{A}^\dagger f^*(x) - \beta^\top \mathcal{A}^\dagger f(x)\right)^2\right]$$

$$\leq \mathbb{E}_x\left[\|\beta\|_2^2 \left\|\mathcal{A}^\dagger f^*(x) - \mathcal{A}^\dagger f(x)\right\|_2^2\right]$$

$$\leq \|\beta\|_2^2 \mathbb{E}_x\left[\left\|\mathcal{A}^\dagger f^*(x) - \mathcal{A}^\dagger f(x)\right\|_1^2\right].$$

We are going to use Lemma A.4 to bound $\left\| \mathcal{A}^\dagger f^*(x) - \mathcal{A}^\dagger f(x) \right\|_1$. By Lemma A.3 we know $f^*(x)$ is the word posterior, so $f^*(x) = \mathbb{E}_{y|x}[y|x]$. Since $\ell$ is cross-entropy loss, we know

$$\mathbb{E}_{y|x}[\ell(f(x), y) - \ell(f^*(x), y)|x] = \mathbb{E}_{y|x}\left[ \sum_{k \in [V^t]} y_k \log\left( \frac{[f^*(x)]_k}{[f(x)]_k} \right) \middle| x \right] = D_{KL}(f^*(x) \parallel f(x)),$$

which implies $\ell(f(x), f^*(x)) - \min_p \ell(p, f^*(x)) \le \epsilon$. Therefore, by Lemma A.4 with $B = \mathcal{A}^\dagger$, we know

$$\left\| \mathcal{A}^\dagger f(x) - \mathcal{A}^\dagger f^*(x) \right\|_1 \le \kappa(\mathcal{A}^\dagger)\sqrt{2\epsilon} = \kappa^t(A^\dagger)\sqrt{2\epsilon}.$$

Given the desired bound, we have

$$\mathbb{E}_x\left[ \left( \mathbb{E}_w[P(w)|x] - \theta^\top f(x) \right)^2 \right] \le 2\|\beta\|_2^2 \kappa^{2t}(A^\dagger)\epsilon.$$

$\square$

## A.4 DISCUSSION ON THE EFFECTIVE DIMENSION OF THE REPRESENTATION

Although the dimension of $f(x)$ is $V^t$, its "effective dimension" is actually $K^t$, which is much smaller than $V^t$ (the number of topic $K$ usually is much smaller than the size of vocabulary $V$). To see this, let's consider the simple case of $t = 1$. As we discussed in the paragraph below Theorem 3.1, we have $f(x) = A\mathbb{E}[w|x, A]$, where $A \in \mathbb{R}^{V \times K}$ is the topic-word matrix and $\mathbb{E}[w|x, A]$ is the posterior mean of $w \in \mathbb{R}^K$. In words, this means that $f(x)$ is completely determined by a linear transformation of a $K$-dimensional vector. Therefore, when we consider the feature of $n$ documents as a matrix $[f(x^{(1)}), ..., f(x^{(n)})] \in \mathbb{R}^{V \times n}$, the rank of this matrix is at most $K$ and one can get these $K$-dimensional features by running standard dimension reduction method (e.g., SVD) on the original feature matrix.

## B OMITTED PROOFS IN SECTION 4

In this section, we give the omitted proofs in Section 4. We give the proof of Theorem 4.1 in Section B.1 and proof of Lemma B.1 in Section B.2.

### B.1 PROOF OF THEOREM 4.1

Before presenting the proof of Theorem 4.1, we first give a characterization of the representation $g(x, x')$, which would be useful in the later analysis. Note that this the same as the one shown in Tosh et al. (2021a). We provide its proof for completeness in Section B.2.

**Lemma B.1.** *If $f$ minimizes the contrastive objective* (2)*, then we have*

$$g(x, x') \triangleq \frac{f(x, x')}{1 - f(x, x')} = \frac{\mathbb{P}(y = 1|x, x')}{\mathbb{P}(y = 0|x, x')}.$$

Now we are ready to proof Theorem 4.1.

**Theorem 4.1.** *Consider the general topic model setting as Definition 2.1, suppose topic-word matrix $A$ satisfies rank$(A) = K$ and function $f$ minimizes the contrastive objective* (2)*. Assume we randomly sampled $m = K^t$ different landmark documents $\{l_i\}_{i=1}^m$ and construct $g(x, \{l_i\}_{i=1}^m)$ as* (3)*. Then, for any document $x$ and its posterior $w|(x, A)$, the posterior mean of any polynomial $P(w)$ with degree at most $t$ is linear in $g(x, \{l_i\}_{i=1}^m)$. That is, there exists $\theta \in \mathbb{R}^m$ such that for all documents $x$*

$$\mathbb{E}_w[P(w)|x, A] = \theta^\top g(x, \{l_i\}_{i=1}^m).$$

*Proof.* Since $f$ is the minimizer of contrastive objective (2), by Lemma B.1 we know

$$g(x, x') = \frac{\mathbb{P}(y = 1|x, x')}{\mathbb{P}(y = 0|x, x')}.$$

Similar to the proof of Theorem 3.1, we will show that $g(x, x')$ is linear in the topic posterior when $x'$ is fixed. By Bayes' rule, we have

$$g(x, x') = \frac{\mathbb{P}(x, x'|y = 1)\mathbb{P}(y = 1)/\mathbb{P}(x, x')}{\mathbb{P}(x, x'|y = 0)\mathbb{P}(y = 0)/\mathbb{P}(x, x')} = \frac{\mathbb{P}(x, x'|y = 1)}{\mathbb{P}(x, x'|y = 0)},$$

where we use $\mathbb{P}(y = 0) = \mathbb{P}(y = 1) = 1/2$.

Denote the $t$ words in $x'$ as $x'_1, \ldots, x'_t$ and their corresponding topics as $z_1, \ldots, z_t$. Then, by our way of generating $x, x'$, we know

$$\mathbb{P}(x, x'|y = 0) = \mathbb{P}(x)\mathbb{P}(x'),$$

$$\mathbb{P}(x, x'|y = 1) = \int_w \mathbb{P}(x'_1, \ldots, x'_t|w, x, y = 1)\mathbb{P}(w, x|y = 1)\mathrm{d}w$$

$$= \int_w \sum_{z_1, \ldots, z_t \in [K]} \mathbb{P}(x'_1, \ldots, x'_t|z_1, \ldots, z_t, w)\mathbb{P}(z_1, \ldots, z_t|w)\mathbb{P}(w, x)\mathrm{d}w,$$

which implies

$$g(x, x') = \frac{1}{\mathbb{P}(x')} \int_w \sum_{z_1, \ldots, z_t \in [K]} \mathbb{P}(x'_1, \ldots, x'_t|z_1, \ldots, z_t)\mathbb{P}(z_1, \ldots, z_t|w)\mathbb{P}(w|x)\mathrm{d}w$$

Note that

$$\mathbb{P}(x'_1, \ldots, x'_t|z_1, \ldots, z_t) = \prod_{i=1}^t \mathbb{P}(x'_i|z_i) = \prod_{i=1}^t A_{x'_1, z_i},$$

where $A$ is the topic-word matrix. Hence,

$$g(x, x') = \frac{1}{\mathbb{P}(x')} \sum_{z_1, \ldots, z_t \in [K]} \prod_{i=1}^{t} A_{x'_1, z_i} \int_w \mathbb{P}(z_1, \ldots, z_t | w) \mathbb{P}(w|x) \mathrm{d}w$$

$$= \frac{1}{\mathbb{P}(x')} \sum_{z_1, \ldots, z_t \in [K]} \prod_{i=1}^{t} A_{x'_1, z_i} \int_w \prod_{i=1}^{t} w_{z_i} \mathbb{P}(w|x) \mathrm{d}w$$

$$= \frac{1}{\mathbb{P}(x')} \sum_{z_1, \ldots, z_t \in [K]} \prod_{i=1}^{t} A_{x'_1, z_i} \mathbb{E}_w \left[ \prod_{i=1}^{t} w_{z_i} \Big| x \right].$$

Recall that the topic posterior tensor is $W_{post} = \mathbb{E}_w[w^{\otimes t}|x] \in \mathbb{R}^{K \times \ldots \times K}$, where each entry $[W_{post}]_{z_1, \ldots, z_k} = \mathbb{E}_w[w_{z_1} \ldots w_{z_t}|x]$. Therefore,

$$g(x, x') = \frac{1}{\mathbb{P}(x')} \sum_{z_1, z_2, \ldots z_t \in [K]} \prod_{i=1}^{t} A_{x'_i, z_i} [W_{post}]_{z_1, z_2, \ldots, z_t}$$

$$= \frac{1}{\mathbb{P}(x')} \mathcal{A}[x']^\top \mathrm{vec}(W_{post}),$$

where $\mathcal{A} = A \otimes A \otimes \cdots \otimes A \in \mathbb{R}^{V^t \times K^t}$, $\mathcal{A}[x'] \in \mathbb{R}^{K^t}$ is the row of $\mathcal{A}$ that correspond to $x'_1, \ldots, x'_t$, and $\mathrm{vec}(W_{post}) \in \mathbb{R}^{K^t}$ is the vectorization of $W_{post}$.

Recall we have $m$ randomly sampled different documents $\{l_i\}_{i=1}^n$. Denote $D \in \mathbb{R}^{m \times m}$ as a diagonal matrix such that $D_{i,i} = \mathbb{P}(l_i)$, and $\tilde{\mathcal{A}} \in \mathbb{R}^{m \times K^t}$ as a submatrix of $\mathcal{A}$ such that each row of $\tilde{\mathcal{A}}$ is $\mathcal{A}[l_i]$. Since $l_i$ are randomly sampled, so $D_{i,i} > 0$. Thus, we know

$$g(x, \{l_i\}_{i=1}^n) = D^{-1} \tilde{\mathcal{A}} \mathrm{vec}(W_{post}),$$

which implies $\mathrm{vec}(W_{post}) = \tilde{\mathcal{A}}^\dagger D g(x, \{l_i\}_{i=1}^n)$. Note that we need to show $\tilde{\mathcal{A}}^\dagger$ is well-defined. Since $A$ has full column rank, we know $\mathcal{A}$ also has full column rank. Thus, the submatrix $\tilde{\mathcal{A}}$ also has full column rank since $m = K^t$. This implies $\tilde{\mathcal{A}}^\dagger$ is well-defined.

Since $P(w)$ is a polynomial of degree at most $t$, we know there exists $\beta$ such that $\mathbb{E}_w[P(w)|x] = \beta^\top \mathrm{vec}(W_{post})$. Thus, let $\theta = D(\tilde{\mathcal{A}}^\dagger)^\top \beta$, we have

$$\mathbb{E}_w[P(w)|x] = \beta^\top \mathrm{vec}(W_{post}) = \theta^\top g(x, \{l_i\}_{i=1}^n).$$

□

## B.2 PROOF OF LEMMA B.1

**Lemma B.1.** *If $f$ minimizes the contrastive objective* (2)*, then we have*

$$g(x, x') \triangleq \frac{f(x, x')}{1 - f(x, x')} = \frac{\mathbb{P}(y = 1|x, x')}{\mathbb{P}(y = 0|x, x')}.$$

*Proof.* Since $f$ is the minimizer of contrastive objective (2) and

$$L_{\mathrm{contrast}}(f) = \mathbb{E}_{x, x', y} \left[ (f(x, x') - y)^2 \right] = \mathbb{E}_{x, x'} \left[ \mathbb{E}_{y|x, x'} \left[ (f(x, x') - y)^2 \right] \right],$$

it is easy to see that $f$ is the Bayes optimal predictor $\mathbb{P}(y = 1|x, x')$. Therefore, we know

$$g(x, x') = \frac{\mathbb{P}(y = 1|x, x')}{\mathbb{P}(y = 0|x, x')}.$$

□

## B.3 Discussions on the Self-referencing

The following corollary shows that solving the downstream task is equivalent to solve a kernel regression (in some sense the landmark documents are just random features for this kernel (Rahimi & Recht, 2007)).

**Corollary B.2.** *Denote $\{(x_i, \tilde{y}_i)\}_{i=1}^m$ as the downstream dataset, where $\tilde{y}_i = \mathbb{E}_w[P(w)|x_i]$ is the target for document $x_i$. If we set landmarks $\{l_i\}_{i=1}^m$ to be $\{x_i\}_{i=1}^m$, then solving the downstream task is equivalent to a kernel regression, that is*

$$\arg\min_\theta \sum_{k=1}^m \left(\tilde{y}_k - \theta^\top g(x_k, \{l_i\}_{i=1}^m)\right)^2 = \arg\min_\theta \|\tilde{y} - G\theta\|^2,$$

*where $\tilde{y} = (\tilde{y}_1, \ldots, \tilde{y}_m)^\top$ and $G \in \mathbb{R}^{m \times m}$ is a kernel matrix such that $G_{ij} = g(x_i, x_j)$.*

As discussed in Corollary B.2, we show that one can use the same set of documents for both representation and downstream task. Recall that in Theorem 4.1 we need a large number of landmark documents to generate the representation and then use it for the downstream task (e.g., fit a polynomial of topic posterior). However, one may not have enough additional documents to be used as landmarks. Therefore, the benefit of the self-referencing is that we do not need additional landmark documents to generate the representation.

The key observation in Corollary B.2 is that the learned representation $g(x, x')$ can be viewed as a kernel. Suppose the documents for the downstream task are $x_1, \ldots, x_m$. Construct a kernel matrix $G \in \mathbb{R}^{m \times m}$ such that $G_{ij} = g(x_i, x_j)$. Then, it is easy to see

$$\theta^\top g(x_i, \{x_j\}_{j=1}^m) = \theta^\top G_i,$$

where $G_i$ is the $i$-th column of $G$. Thus, $\theta$ can be found by the following kernel regression:

$$\min_\theta \|\tilde{y} - G\theta\|^2,$$

where $\tilde{y} = (\tilde{y}_1, \ldots, \tilde{y}_m)^\top$ and $\tilde{y}_i = \mathbb{E}_w[P(w)|x_i]$.

## C SYNTHETIC EXPERIMENT ADDITIONAL DETAILS

In this section, we include more details about our synthetic experiments in Section 5. In Section C.1, we describe the document generation process for PAM model. Additional results on topic posterior recovery loss and major topic recovery are included in Section C.2 and Section C.3. In Section C.4 and Section C.5, we report the results about different training epochs and hyperparameters tuning. $\ell_1$ condition number is reported in Section C.6 and we give some visualizations for topic correlations for the PAM model in Section C.7. Moreover, in Section C.8 we report the performance of SSL on recovering the topic posterior mean vector when $t = 2$. We run our experiments on 2080RTXTi/V100s. The confidence interval reported in the paper is calculated as $mean \pm 1.96 * s_{testdocs}/\sqrt{N_{testdocs}}$, where $mean$ is the mean of TV distance (or accuracy), $s_{testdocs}$ is the standard deviation of TV distance (or accuracy) and $N_{testdocs}$ is the number of test documents.

**Details of generating topic-word matrix** $A$   For pure topic and LDA experiments, each column of $A$ (i.e., each topic's word distribution) is generated from $Dir(\alpha/K)$. For CTM and PAM, our goal is to construct $K/4$ groups of 4 topics such that within each group, topic 0 and topic 1 have similar word distribution, and topic 2 and topic 3 have different word distribution (and different from the word distribution of topic 0 and 1 as well). To achieve this, we first sample $K/4$ word distribution vectors from $Dir(\alpha/K)$, and for each word distribution vector, we permute its entries to get the word distribution of topic 0,1,2,3 within each group and intentionally align the large entries in topic 0 and 1 to ensure high similarity between them.

### C.1 PACHINKO ALLOCATION MODEL TOPIC PRIOR GENERATION

The topic proportion of the Pachinko Allocation Model (PAM) described in Section 5 is generated from the following process: for each document, we first sample a "super-topic" proportion from a symmetric Dirichlet distribution $\text{Dir}(1/K_s)$ and a super-topic-to-topic proportion from a symmetric Dirichlet distribution $\text{Dir}(30)$. Then, we sample each word by first sample a super-topic according to the super-topic proportion and then sample the actual topic from the super-topic-to-topic proportion. In our experiment, we set $K_s = 10$. Figure C.1 offers a few examples of PAM topic proportion generated from the PAM model.

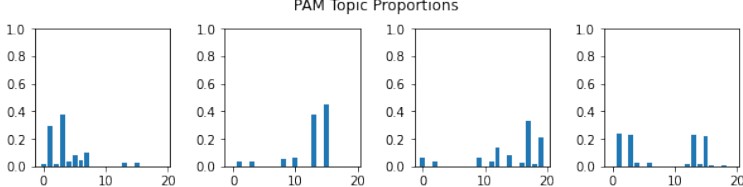

Figure C.1: PAM topic proportion

### C.2 MAJOR TOPIC RECOVERY ADDITIONAL RESULTS

In Section 5.3, we measure the major topics recovery of CTM and PAM documents as the top-2 topic overlap rate. Since topics are correlated in pairs in both of these topic models, we are also interested in the top-4 and top-6 topics overlap rate. As shown in Figure C.2, for both CTM and PAM documents, the average top-4 overlap rate is above 66% and the average top-6 overlap rate is above 62% on all $\alpha$ values we test on. These results show that our model can accurately capture more than just the major pair of correlated topics, but also some correlated pairs of less dominant topics.

### C.3 SELF-SUPERVISED LEARNING VERSUS ASSUMING SPECIFIC TOPIC PRIOR

We present in Section 5.3 a comparison between the performance of our model and Markov Chain Monte Carlo assuming a specific topic model for $\alpha = 1$, on the task of TV distance and major topic recovery. In this subsection, we supplement the experiment results for larger values of $\alpha$ along with $\alpha = 1$. In particular, we consider self-supervised learning with 6 million training documents and

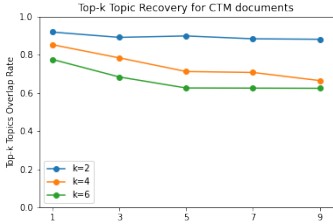 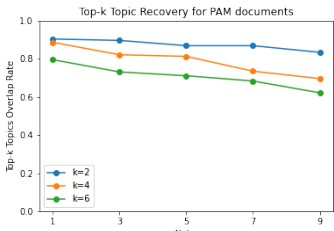

Figure C.2: Recovery rate for top-2, top-4, and top-6 topics, measured on CTM (left) and PAM (right) test documents.

| Method | Pure | Document Type ($\alpha = 1$) LDA | CTM | PAM |
|---|---|---|---|---|
| LDA | $0.0406 \pm 0.0016$ | - | $0.1182 \pm 0.0099$ | $0.1218 \pm 0.0096$ |
| CTM | $0.2083 \pm 0.0038$ | $0.2060 \pm 0.0064$ | - | $0.3154 \pm 0.0082$ |
| PAM | $0.3782 \pm 0.0038$ | $0.3459 \pm 0.0128$ | $0.3939 \pm 0.0096$ | - |
| **SSL-6M (ours)** | **0.0148 ± 0.0020** | **0.0757 ± 0.0033** | **0.0550 ± 0.0037** | **0.0489 ± 0.0025** |
| **SSL-120K (ours)** | $0.0311 \pm 0.0025$ | $0.0878 \pm 0.0041$ | $0.0678 \pm 0.0038$ | $0.0600 \pm 0.0028$ |
| Method | Pure | Document Type ($\alpha = 3$) LDA | CTM | PAM |
| LDA | $0.0561 \pm 0.0030$ | - | $0.1942 \pm 0.0156$ | $0.1813 \pm 0.0146$ |
| CTM | $0.2347 \pm 0.0048$ | $0.2299 \pm 0.0082$ | - | $0.3362 \pm 0.0086$ |
| PAM | $0.4031 \pm 0.0046$ | $0.3665 \pm 0.0117$ | $0.4510 \pm 0.0100$ | - |
| **SSL-6M (ours)** | **0.0308 ± 0.0070** | **0.0899 ± 0.0053** | **0.0799 ± 0.0048** | **0.0712 ± 0.0038** |
| **SSL-120K (ours)** | $0.0546 \pm 0.0097$ | $0.1292 \pm 0.0067$ | $0.1062 \pm 0.0058$ | $0.0988 \pm 0.0048$ |
| Method | Pure | Document Type ($\alpha = 5$) LDA | CTM | PAM |
| LDA | $0.0731 \pm 0.0040$ | - | $0.2123 \pm 0.0159$ | $0.1995 \pm 0.0144$ |
| CTM | $0.2655 \pm 0.0059$ | $0.2413 \pm 0.0090$ | - | $0.3428 \pm 0.0090$ |
| PAM | $0.4337 \pm 0.0055$ | $0.3608 \pm 0.0089$ | $0.4794 \pm 0.0097$ | - |
| **SSL-6M (ours)** | **0.0501 ± 0.0030** | **0.1041 ± 0.0062** | **0.0970 ± 0.0057** | **0.0787 ± 0.0043** |
| **SSL-120K (ours)** | $0.0832 \pm 0.0174$ | $0.1629 \pm 0.0082$ | $0.1245 \pm 0.0071$ | $0.1077 \pm 0.0054$ |
| Method | Pure | Document Type ($\alpha = 7$) LDA | CTM | PAM |
| LDA | $0.0883 \pm 0.0050$ | - | $0.2438 \pm 0.0169$ | $0.2138 \pm 0.0158$ |
| CTM | $0.2840 \pm 0.0060$ | $0.2556 \pm 0.0089$ | - | $0.3530 \pm 0.0099$ |
| PAM | $0.4561 \pm 0.0052$ | $0.3707 \pm 0.0089$ | $0.4979 \pm 0.0101$ | - |
| **SSL-6M (ours)** | **0.0391 ± 0.0022** | **0.1233 ± 0.0071** | **0.1071 ± 0.0068** | **0.0960 ± 0.0057** |
| **SSL-120K (ours)** | $0.1219 \pm 0.0227$ | $0.1851 \pm 0.0079$ | $0.1340 \pm 0.0078$ | $0.1358 \pm 0.0070$ |
| Method | Pure | Document Type ($\alpha = 9$) LDA | CTM | PAM |
| LDA | $0.1051 \pm 0.0065$ | - | $0.2559 \pm 0.0162$ | $0.2281 \pm 0.0152$ |
| CTM | $0.3129 \pm 0.0079$ | $0.2580 \pm 0.0092$ | - | $0.3556 \pm 0.0084$ |
| PAM | $0.4805 \pm 0.0071$ | $0.3776 \pm 0.0093$ | $0.5156 \pm 0.0088$ | - |
| **SSL-6M (ours)** | **0.0517 ± 0.0045** | **0.1358 ± 0.0089** | **0.1101 ± 0.0064** | **0.0971 ± 0.0053** |
| **SSL-120K (ours)** | $0.1121 \pm 0.0202$ | $0.2221 \pm 0.0123$ | $0.1449 \pm 0.0077$ | $0.1460 \pm 0.0070$ |

Table C.1: TV between our recovered topic posterior and the true topic posterior of our self-supervised learning approach versus topic inference via Markov Chain Monte Carlo assuming a specific prior for $\alpha = 1, 3, 5, 7, 9$. The 95% confidence interval is reported.

with 120 thousand training documents after 200 training epochs, and we expect that the former can allow our model to approximate the optimal predictor $f$ described in Theorem 3.1 and the latter may offer some insights on our approach's performance in more practical settings, such as on a real world dataset with limited size. Additionally, when training with 120 thousand documents, we randomly shuffle each document and reassign the $t$ target words in every epoch, to make sure our model learns enough information from the limited amount of training data. We report our results in the form of 95% confidence interval.

Table C.1 shows that our approach with 6 million training documents consistently outperforms misspecified topic priors by yielding a much lower TV distance from the true topic posterior. Note that the entries with correct topic model are omitted, because in our experiment the Markov Chain Monte Carlo recovered topic posterior assuming the correct topic model is exactly what we use as

| | Document Type ($\alpha = 1$) | | |
|---|---|---|---|
| Method | Pure | LDA | CTM | PAM |
| LDA | **1.0 ± 0.0** | **0.9150 ± 0.0387** | 0.8850 ± 0.0344 | 0.8500 ± 0.0360 |
| CTM | **1.0 ± 0.0** | 0.9000 ± 0.0416 | **0.9175 ± 0.0275** | 0.8300 ± 0.0370 |
| PAM | **1.0 ± 0.0** | 0.8750 ± 0.0458 | 0.7250 ± 0.0397 | 0.9050 ± 0.0320 |
| **SSL-6M (ours)** | **1.0 ± 0.0** | 0.9050 ± 0.0406 | **0.9175 ± 0.0275** | 0.9025 ± 0.0330 |
| **SSL-120K (ours)** | **1.0 ± 0.0** | **0.9150 ± 0.0387** | 0.9100 ± 0.0292 | **0.9200 ± 0.0289** |

| | Document Type ($\alpha = 3$) | | |
|---|---|---|---|
| Method | Pure | LDA | CTM | PAM |
| LDA | **1.0 ± 0.0** | **0.8900 ± 0.0434** | 0.8200 ± 0.0354 | 0.8200 ± 0.0404 |
| CTM | **1.0 ± 0.0** | 0.8800 ± 0.0450 | **0.8900 ± 0.0326** | 0.7650 ± 0.0404 |
| PAM | **1.0 ± 0.0** | 0.8550 ± 0.0488 | 0.6375 ± 0.0392 | **0.8950 ± 0.0357** |
| **SSL-6M (ours)** | **1.0 ± 0.0** | 0.8750 ± 0.0458 | **0.8900 ± 0.0311** | **0.8950 ± 0.0344** |
| **SSL-120K (ours)** | 0.9950 ± 0.0098 | 0.8700 ± 0.0466 | **0.8900 ± 0.0312** | **0.8950 ± 0.0357** |

| | Document Type ($\alpha = 5$) | | |
|---|---|---|---|
| Method | Pure | LDA | CTM | PAM |
| LDA | **1.0 ± 0.0** | **0.9050 ± 0.0406** | 0.8500 ± 0.0339 | 0.7750 ± 0.0390 |
| CTM | **1.0 ± 0.0** | **0.9050 ± 0.0406** | 0.8975 ± 0.0304 | 0.6775 ± 0.0421 |
| PAM | **1.0 ± 0.0** | 0.8850 ± 0.0442 | 0.6150 ± 0.0371 | **0.8825 ± 0.0379** |
| **SSL-6M (ours)** | **1.0 ± 0.0** | 0.9000 ± 0.0416 | **0.8975 ± 0.0296** | 0.8675 ± 0.0407 |
| **SSL-120K (ours)** | 0.9900 ± 0.0138 | 0.8800 ± 0.0450 | 0.8900 ± 0.0303 | 0.8750 ± 0.0390 |

| | Document Type ($\alpha = 7$) | | |
|---|---|---|---|
| Method | Pure | LDA | CTM | PAM |
| LDA | **1.0 ± 0.0** | 0.8800 ± 0.0450 | 0.7750 ± 0.0397 | 0.7475 ± 0.0433 |
| CTM | **1.0 ± 0.0** | 0.8650 ± 0.0474 | **0.8825 ± 0.0353** | 0.6250 ± 0.0390 |
| PAM | **1.0 ± 0.0** | 0.8450 ± 0.0502 | 0.5725 ± 0.0356 | 0.8700 ± 0.0411 |
| **SSL-6M (ours)** | **1.0 ± 0.0** | 0.9150 ± 0.0387 | 0.8675 ± 0.0383 | 0.8675 ± 0.0407 |
| **SSL-120K (ours)** | 0.9650 ± 0.0255 | **0.9350 ± 0.0342** | 0.8775 ± 0.0370 | **0.8725 ± 0.0398** |

| | Document Type ($\alpha = 9$) | | |
|---|---|---|---|
| Method | Pure | LDA | CTM | PAM |
| LDA | **1.0 ± 0.0** | 0.8850 ± 0.0442 | 0.7750 ± 0.0384 | 0.7350 ± 0.0432 |
| CTM | **1.0 ± 0.0** | 0.8950 ± 0.0425 | 0.8800 ± 0.0348 | 0.5925 ± 0.0392 |
| PAM | **1.0 ± 0.0** | 0.8650 ± 0.0474 | 0.5675 ± 0.0358 | **0.8600 ± 0.0428** |
| **SSL-6M (ours)** | 0.9950 ± 0.0098 | **0.9100 ± 0.0397** | **0.8850 ± 0.0330** | 0.8325 ± 0.0461 |
| **SSL-120K (ours)** | 0.9800 ± 0.0194 | 0.8600 ± 0.0481 | 0.8800 ± 0.0341 | 0.8425 ± 0.0430 |

Table C.2: Major topic recovery rate of our approach versus posterior inference via Markov Chain Monte Carlo assuming a specific prior for $\alpha = 1, 3, 5, 7, 9$. We report the 95% confidence interval.

our ground truth topic posterior. Meanwhile, our approach with 120 thousand training documents outperforms misspecified prior in almost every scenario, except when compared to LDA prior on pure topic documents. This is is likely because the LDA topic prior is relatively concentrated on one specific topic and thus similar to the pure topic prior.

We also present a comparison between our self-supervised learning approach and posterior inference assuming a specific topic prior on the task of recovering major topics in the topic proportion for $\alpha = 1, 3, 5, 7, 9$, as a supplement to Section 5.3. As shown in Table C.2, in almost every scenario we test on, our model can perform competitively against the MCMC-inferred posterior assuming the correct topic prior and outperform misspecified topic prior.

To further investigate the performance of self-supervised learning, we compare our approach with Variational Inference assuming a specific topic prior on the task of recovering major topics in test document's topic proportion. Table C.3 shows that, similar to our findings from Table C.2, our model can perform competitively against the Variational Inference posterior assuming the correct topic prior, particularly for relative small $\alpha$ values, and can outperform misspecified topic prior in almost every test scenario.

## C.4 THE EFFECT OF TRAINING EPOCHS

We measure how the number of training epochs may influence topic posterior recovery loss by varying the number of training epochs. The results we present in Section 5 are based on models trained for

| | Document Type ($\alpha = 1$) | | | |
|---|---|---|---|---|
| Method | Pure | LDA | CTM | PAM |
| LDA | **1.0 ± 0.0** | 0.8950 ± 0.0424 | 0.8325 ± 0.0382 | 0.8625 ± 0.0339 |
| CTM | **1.0 ± 0.0** | 0.9050 ± 0.0410 | 0.9125 ± 0.0283 | 0.8150 ± 0.0354 |
| PAM | **1.0 ± 0.0** | 0.8600 ± 0.0481 | 0.7125 ± 0.0382 | 0.9050 ± 0.0325 |
| **SSL-6M (ours)** | **1.0 ± 0.0** | 0.9050 ± 0.0406 | **0.9175 ± 0.0275** | 0.9025 ± 0.0330 |
| **SSL-120K (ours)** | **1.0 ± 0.0** | **0.9150 ± 0.0387** | 0.9100 ± 0.0292 | **0.9200 ± 0.0289** |
| | Document Type ($\alpha = 3$) | | | |
| Method | Pure | LDA | CTM | PAM |
| LDA | **1.0 ± 0.0** | 0.8750 ± 0.0452 | 0.7325 ± 0.0396 | 0.7875 ± 0.0382 |
| CTM | **1.0 ± 0.0** | **0.8800 ± 0.0450** | 0.8800 ± 0.0325 | 0.6975 ± 0.0410 |
| PAM | **1.0 ± 0.0** | 0.8550 ± 0.0488 | 0.6075 ± 0.0354 | 0.8825 ± 0.0367 |
| **SSL-6M (ours)** | **1.0 ± 0.0** | 0.8750 ± 0.0458 | **0.8900 ± 0.0311** | **0.8950 ± 0.0344** |
| **SSL-120K (ours)** | 0.9950 ± 0.0098 | 0.8700 ± 0.0466 | **0.8900 ± 0.0312** | **0.8950 ± 0.0357** |
| | Document Type ($\alpha = 5$) | | | |
| Method | Pure | LDA | CTM | PAM |
| LDA | **1.0 ± 0.0** | 0.8800 ± 0.0452 | 0.7575 ± 0.0354 | 0.7375 ± 0.0410 |
| CTM | **1.0 ± 0.0** | 0.8750 ± 0.0452 | **0.8975 ± 0.0283** | 0.5925 ± 0.0396 |
| PAM | **1.0 ± 0.0** | 0.8500 ± 0.0495 | 0.5675 ± 0.0297 | **0.8875 ± 0.0368** |
| **SSL-6M (ours)** | **1.0 ± 0.0** | **0.9000 ± 0.0416** | **0.8975 ± 0.0296** | 0.8675 ± 0.0407 |
| **SSL-120K (ours)** | 0.9900 ± 0.0138 | 0.8800 ± 0.0450 | 0.8900 ± 0.0303 | 0.8750 ± 0.0390 |
| | Document Type ($\alpha = 7$) | | | |
| Method | Pure | LDA | CTM | PAM |
| LDA | **1.0 ± 0.0** | 0.9250 ± 0.0368 | 0.7175 ± 0.0396 | 0.6875 ± 0.0410 |
| CTM | **1.0 ± 0.0** | 0.9300 ± 0.0354 | 0.8700 ± 0.0382 | 0.5700 ± 0.0354 |
| PAM | **1.0 ± 0.0** | 0.9050 ± 0.0410 | 0.5400 ± 0.0325 | **0.8750 ± 0.0396** |
| **SSL-6M (ours)** | **1.0 ± 0.0** | 0.9150 ± 0.0387 | 0.8675 ± 0.0383 | 0.8675 ± 0.0407 |
| **SSL-120K (ours)** | 0.9650 ± 0.0255 | **0.9350 ± 0.0342** | **0.8775 ± 0.0370** | 0.8725 ± 0.0398 |
| | Document Type ($\alpha = 9$) | | | |
| Method | Pure | LDA | CTM | PAM |
| LDA | **1.0 ± 0.0** | 0.9050 ± 0.0410 | 0.7250 ± 0.0396 | 0.6975 ± 0.0410 |
| CTM | **1.0 ± 0.0** | **0.9200 ± 0.0382** | 0.8725 ± 0.0339 | 0.5425 ± 0.0368 |
| PAM | **1.0 ± 0.0** | 0.8750 ± 0.0452 | 0.5300 ± 0.0325 | **0.8425 ± 0.0452** |
| **SSL-6M (ours)** | 0.9950 ± 0.0098 | 0.9100 ± 0.0397 | **0.8850 ± 0.0330** | 0.8325 ± 0.0461 |
| **SSL-120K (ours)** | 0.9800 ± 0.0194 | 0.8600 ± 0.0481 | 0.8800 ± 0.0341 | **0.8425 ± 0.0430** |

Table C.3: Major topic recovery rate of our approach versus posterior inference via Variational Inference assuming a specific prior for $\alpha = 1, 3, 5, 7, 9$. We report the 95% confidence interval.

200 epochs for all types of documents. Here, we present the topic recovery loss, measured as the Total Variation distance between our recovered topic posterior and the true topic posterior, for epochs=25, 50, 100, 200 using the same model architecture and same sampling scheme as in Section 5.

Table C.4 shows that topic posterior recovery loss steadily decreases as the number of epochs gets larger. Interestingly, even with just training 50 epochs, our model can recover the topic posterior within a Total Variation distance of less than 0.3 off the true topic posterior for documents generated from all four topic models.

## C.5 Hyperparameter Tuning Experiments

As described in Section 5, the two types of neural network architecture we use are fully-connected neural networks and attention-based neural networks (see Figure C.3). In this section, we present some of our TV results of different combinations of hyperparameters for both types of models. Note that our attention-based neural network slightly differs from typical transformers (Vaswani et al., 2017) in that we use batch normalization instead of layer normalization in every residual connection. We only use one attention head in each multi-head attention layer. During training, we use AMSGrad optimizer and an initial learning rate of 0.0001 or 0.0002. We reduce the learning rate by 50% whenever validation loss does not decrease in 10 epochs.

The main hyperparameters in our fully-connected neural networks include the hidden dimension, the number of layers, and whether we apply residual connections. Table C.5 shows the TV distance results when we vary these hyperparameters for the LDA scenario when $\alpha = 1$. However, fully-connected

| Document type | Number of epochs | Dirichlet hyperparameter $\alpha$ | | | | |
|---|---|---|---|---|---|---|
| | | 1 | 3 | 5 | 7 | 9 |
| Pure-topic | 25 | 0.0505 | 0.1025 | 0.1756 | 0.6222 | 0.8285 |
| | 50 | 0.0310 | 0.0744 | 0.0991 | 0.1522 | 0.2873 |
| | 100 | 0.0160 | 0.0363 | 0.0511 | 0.0911 | 0.1257 |
| | 200 | 0.0148 | 0.0308 | 0.0501 | 0.0391 | 0.0517 |
| LDA | 25 | 0.1236 | 0.1583 | 0.1812 | 0.2022 | 0.2218 |
| | 50 | 0.0967 | 0.1428 | 0.1387 | 0.1953 | 0.2117 |
| | 100 | 0.0805 | 0.1018 | 0.1101 | 0.1361 | 0.1538 |
| | 200 | 0.0709 | 0.0951 | 0.1045 | 0.1222 | 0.1377 |
| CTM | 25 | 0.1089 | 0.1599 | 0.1563 | 0.1747 | 0.1971 |
| | 50 | 0.0900 | 0.1132 | 0.1425 | 0.1455 | 0.1501 |
| | 100 | 0.0623 | 0.0948 | 0.1102 | 0.1187 | 0.1243 |
| | 200 | 0.0550 | 0.0799 | 0.0970 | 0.1071 | 0.1101 |
| PAM | 25 | 0.1072 | 0.1342 | 0.1480 | 0.1688 | 0.1754 |
| | 50 | 0.0820 | 0.1036 | 0.1246 | 0.1185 | 0.1395 |
| | 100 | 0.0543 | 0.0755 | 0.0859 | 0.1087 | 0.1095 |
| | 200 | 0.0436 | 0.0659 | 0.0787 | 0.0953 | 0.0971 |

Table C.4: Topic posterior recovery loss of our self-supervised learning approach, measured in Total Variation distance, for all four types of documents and $\alpha = 1, 3, 5, 7, 9$. The number of training epochs ranges from 25 to 200, and we sample 60K new training documents in every 2 epochs, which corresponds to 720K to 6M training documents.

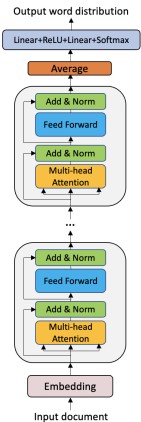

Figure C.3: Our attention-based neural network architecture

neural networks perform poorly in CTM and PAM case. For instance, when $\alpha = 1$ and hidden dimension = 4096, we find that hyperparameters that work well in the LDA case no longer yield satisfactory results (Table C.6).

| TV | Hidden dimension | | |
|---|---|---|---|
| # layers (residual) | 1024 | 2048 | 4096 |
| 3 | 0.1509 | 0.1112 | 0.1095 |
| 6 | 0.7902 | 0.7893 | 0.7901 |
| 3 (residual) | 0.0871 | 0.0867 | 0.0862 |
| 6 (residual) | 0.0822 | 0.0835 | **0.0709** |

Table C.5: Fully-connected neural network's performance in the LDA $\alpha = 1$ scenario.

For attention-based neural networks applied to CTM and PAM scenarios, we mainly consider the number of layers and the hidden dimension per attention layer. Table C.7 presents the TV distance between recovered topic posterior and true topic posterior in the CTM setting, with varying number of layers and attention dimension for $\alpha = 1, 3, 5$ and using a more coarse estimate to the true topic posterior than what we use as our final true CTM topic posterior. We find that the model performs the best when attention dimension is 768 or 1024, and when the attention dimension increases to 2048 the model gives a higher TV (see Table C.8).

| TV | Layer Type | |
| --- | --- | --- |
| # layers | regular | residual |
| 3 | 0.1824 | 0.1665 |
| 4 | 0.1724 | 0.1656 |
| 6 | 0.1700 | 0.1556 |

Table C.6: Fully-connected neural network's performance in the CTM $\alpha = 1$ scenario with 4096 hidden dimensions.

| TV | | | # layers | |
| --- | --- | --- | --- | --- |
| $\alpha$ | Attention dimension | 4 | 6 | 8 |
| 1 | 768 | 0.0902 | 0.0844 | **0.0814** |
| | 1024 | 0.0851 | 0.0890 | 0.0836 |
| 3 | 768 | 0.1471 | 0.1429 | 0.1398 |
| | 1024 | 0.1440 | 0.1384 | **0.1383** |
| 5 | 768 | 0.1794 | 0.1767 | **0.1708** |
| | 1024 | 0.1878 | 0.1787 | 0.1782 |

Table C.7: Attention-based neural network's performance on CTM documents for $\alpha = 1, 3, 5$.

| TV | Attention dimension | | |
| --- | --- | --- | --- |
| $\alpha$ | 768 | 1024 | 2048 |
| 3 | 0.1471 | **0.1440** | 0.1670 |
| 5 | **0.1794** | 0.1878 | 0.1922 |

Table C.8: 4-layer attention-base neural network's performance on CTM documents when attention layer's dimension varies, for $\alpha = 3, 5$.

## C.6 TOPIC POSTERIOR RECOVERY LOSS AND CONDITION NUMBER

We have proved in Theorem 3.3 that the upper bound of topic posterior recovery loss depends on the $\ell_1$ Condition Number $\kappa(A^\dagger)$. In this section, we show that $\kappa(A^\dagger)$ is small for every topic-word matrix $A$ in our experiments (Figure C.4). Note that we use the same topic-word matrix $A$ for pure-topic model and the LDA model. For CTM and PAM, we use the same topic-word matrix up to reordering of the topics that corresponds with pairwise topic correlations. Therefore, pure-topic model and LDA model share the same $\kappa(A^\dagger)$, and CTM and PAM share the same $\kappa(A^\dagger)$.

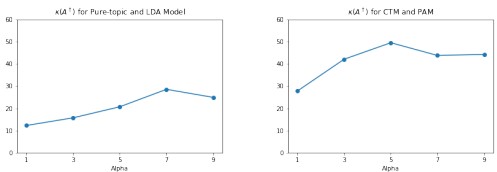

Figure C.4: $\ell_1$ Condition Number for topic-word matrix $A$

## C.7 VISUALIZING TOPIC CORRELATIONS FOR PAM

For documents generated by the PAM model, we plotted the estimated posteriors by the self-supervised approach and posterior inference assuming different priors in Figure C.5. From this figure (especially in Documents 3 and 4), one can qualitatively see that the estimated posterior and the PAM MCMC posteriors are aware of the correlation between topics, while the LDA MCMC posterior fails to take the correlation into consideration and hence is more different from the ground truth topic proportions.

## C.8 TOPIC POSTERIOR RECOVERY FOR $t = 2$

We have shown by experiment that self-supervised learning can accurately recover the topic posterior when $t = 1$, that is, when the neural network model is trained to predict one missing word from a document. In this subsection, we run experiments for $t = 2$ case in Theorem 3.3, which means using the self-supervised objective to train the neural network model to predict 2 missing word in a document. For the sake of reducing computational complexity, we change the setting from

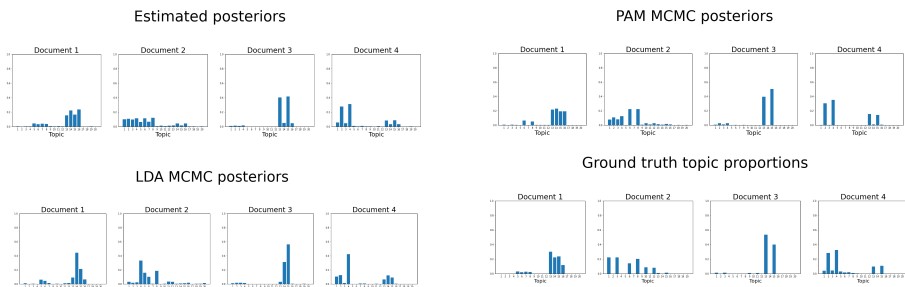

Figure C.5: Comparison of estimated, PAM MCMC, LDA MCMC, and ground truth posterior topic proportions for sample test documents generated by PAM. Overall, across our set of 200 test documents, the estimated posteriors were able to outperform the misspecified LDA posteriors.

$V = 5000, K = 20, K_s = 10$ to $V = 500, K = 8$, and $K_s = 4$ (where $K_s$ is a parameter in the PAM model) while keeping document length, resample rate, training epoch, and the number of test documents the same as in Section 5. Once a neural network $f$ is trained, from the proof of Theorem 3.1 in Section A.2 we know that the training loss minimizer $f$ satisfies $f(x) = (A \otimes A)\text{vec}(W_{post})$ when $t = 2$, so we obtain $\text{vec}(W_{post})$ by $\text{vec}(W_{post}) = (A \otimes A)^\dagger f(x)$ where we expect $W_{post}$ to be a symmetric matrix. But we find that in our experiment $W_{post}$ is nearly symmetric but not perfectly symmetric, and to amend this, we use $\frac{1}{2}(W_{post} + W_{post}^T)$ as the SSL-recovered posterior tensor.

|  | Document Type | | | |
|---|---|---|---|---|
| $\alpha$ | Pure | LDA | CTM | PAM |
| 1 | 8 | 8 | 6 | 6 |
| 3 | 8 | 8 | 6 | 6 |
| 5 | 8 | 8 | 6 | 6 |
| 7 | 8 | 8 | 8 | 6 |
| 9 | 8 | 8 | 8 | 10 |

Table C.9: The number of attention blocks used in the neural network architecture for $t = 2$ for all four types of documents and $\alpha = 1, 3, 5, 7, 9$.

We observe that the attention-based neural network architecture with 384 hidden dimension and varying number of attention blocks (shown in Table C.9) is performs better on learning the distribution of 2 missing words than the fully-connected neural network for all four types of documents, so all of the following results are based on the output of trained attention-based neural network. We use the posterior mean from MCMC assuming the correct prior as the ground-truth topic posterior, and use the TV distance to measure how far the SSL-recovered topic posterior deviates from it, as shown in Table C.10. In most scenarios, the TV distance is less than 0.15, and TV distance gets slightly larger as $\alpha$ increases, i.e., the word distribution of each topic gets less distinguishable.

| TV Distance | | Document Type | | |
|---|---|---|---|---|
| $\alpha$ | Pure | LDA | CTM | PAM |
| 1 | $0.0081 \pm 0.0002$ | $0.0636 \pm 0.0031$ | $0.0748 \pm 0.0053$ | $0.0748 \pm 0.0035$ |
| 3 | $0.0291 \pm 0.0003$ | $0.0870 \pm 0.0039$ | $0.1206 \pm 0.0069$ | $0.1045 \pm 0.0047$ |
| 5 | $0.0431 \pm 0.0004$ | $0.1075 \pm 0.0050$ | $0.1523 \pm 0.0067$ | $0.1544 \pm 0.0062$ |
| 7 | $0.0578 \pm 0.00004$ | $0.1258 \pm 0.0049$ | $0.1627 \pm 0.0058$ | $0.1693 \pm 0.0065$ |
| 9 | $0.0662 \pm 0.0006$ | $0.1413 \pm 0.0052$ | $0.1859 \pm 0.0061$ | $0.2770 \pm 0.0152$ |

Table C.10: Topic posterior recovery loss of our self-supervised learning approach for $t = 2$, measured in Total Variation distance, for all four types of documents and $\alpha = 1, 3, 5, 7, 9$. We report the 95% confidence interval.

We also measure how the largest entries in the SSL-recovered topic posterior overlap with the largest entries in the topic prior. Note that for each document, since the underlying topic of each word is

| | | Document Type | | |
|---|---|---|---|---|
| $\alpha$ | Pure | LDA | CTM | PAM |
| 1 | $1.0 \pm 0.0$ | $0.9350 \pm 0.0342$ | $0.9088 \pm 0.0286$ | $0.8638 \pm 0.0402$ |
| 3 | $1.0 \pm 0.0$ | $0.8800 \pm 0.0450$ | $0.8700 \pm 0.0363$ | $0.8075 \pm 0.0476$ |
| 5 | $1.0 \pm 0.0$ | $0.8950 \pm 0.0425$ | $0.8050 \pm 0.0473$ | $0.8012 \pm 0.0512$ |
| 7 | $1.0 \pm 0.0$ | $0.9100 \pm 0.0397$ | $0.8450 \pm 0.0428$ | $0.8125 \pm 0.0472$ |
| 9 | $1.0 \pm 0.0$ | $0.8950 \pm 0.0425$ | $0.7675 \pm 0.0505$ | $0.6625 \pm 0.0485$ |

Table C.11: The major topic pair recovery rate of our self-supervised learning approach for $t = 2$ for all four types of documents and $\alpha = 1, 3, 5, 7, 9$. We report the 95% confidence interval.

drawn iid, the joint topic prior distribution for the 2 missing words is the Kronecker product of the document's topic prior distribution with itself. For the overlap rate, in the $t = 1$ case we measured the overlap of the top 1 topic for pure topic and LDA documents and the overlap of top 2 topics for CTM and PAM documents, because most pure topic documents and LDA documents are generated by one dominant topic whereas most CTM and PAM documents are generated by two correlated dominant topics. Similarly, in the $t = 2$ case we measure the overlap of the top 1 entry for pure topic and LDA documents and the overlap of top $2 \times 2 = 4$ entries for CTM and PAM documents. We will refer to this overlap rate as the "major topic pair recovery rate". The results are shown in Table C.11. Overall, the overlap rate is very high, which indicates that for $t = 2$ our self-supervised learning approach can capture the most dominant topic(s) for the 2 missing words.

## C.9   DETAILS OF MCMC NUTS SAMPLING METHOD

In our experiments, we use PyMC3 package Salvatier et al. (2016) to perform NUTS-based MCMC posterior sampling. Specifically, we use 3000 tuning iterations to tune the sampler's step size so that we approximate an acceptance rate of 0.9. After tuning, we draw 2000 samples from the posterior distribution and use the mean of these 2000 samples as our best estimate for the topic posterior mean.

To further check the performance of NUTS, we look into the LDA documents. As shown below, we observe that MCMC posterior assuming LDA model (bottom row) is also very similar to the ground truth $w$ (top row) used to generate the documents.

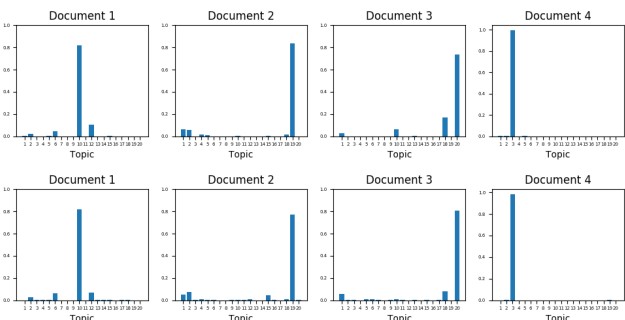

Figure C.6: Comparison of NUTS-based MCMC topic posterior assuming correct model and the document-generating topic proportions $w$ for sample test documents generated by LDA.

## D    REAL-DATA EXPERIMENT ADDITIONAL DETAILS

In this section, we give more details about our real data experiments in Section 6. In Section D.1, we describe the data processing. We give a more detailed description of baselines and how we extracted our representations in Section D.2. In Section D.3, we report more results using different model architectures. Moreover, in Section  D.4, we offer details about experiments on other real-world datasets and report the results.

### D.1    DATA PROCESSING

Here we detailed our usage of the AG news dataset by Zhang et al. (2015). Each category has 30,000 samples in the training set and 19,000 samples in the testing set. We first preprocessed the data by removing punctuation and words that occurred in fewer than 10 documents, obtaining in a vocabulary of around 16,700 words, in a similar fashion done by Tosh et al. (2021b).

To split the data set into unsupervised dataset and supervised dataset, we selected a random sample of 1000 documents as labeled supervised dataset for each category of documents, while the remaining 116,000 documents fall into unsupervised dataset for representation learning.

### D.2    EXTRACTING REPRESENTATIONS

To give some details about two baseline representations we used, we described in details as follows:

- **Bag of Words (BOW):** For a single document, we constructs BOW embedding by creating a bag-of-words frequency vector of the dimension of vocabulary size, where each entry $i$ represent the frequency of words with id $i$ in that particular document.
- **Word2vec:** To generate Word2vec representation fitted the Skip-gram word embedding model on unsupervised dataset Mikolov et al. (2013). The implementation is done through the Gensim library Rehurek & Sojka (2011), where we used an embedding dimension of 200 (after tuning, see Table D.1) and window size of 5. Representation is taken as the average of all trained word embeddings in a single document.
- **LDA:** We construct a representation derived from LDA by first fitting Latent Dirichlet Allocation model Blei et al. (2003) on the unsupervised dataset and then use the posterior distribution over topics given a document as representation. We implemented LDA using scikit-learn Pedregosa et al. (2011) with default parameters except for number of topics, which we use as representation size. We use final dimension of 100 (after tuning, see Figure D.1).

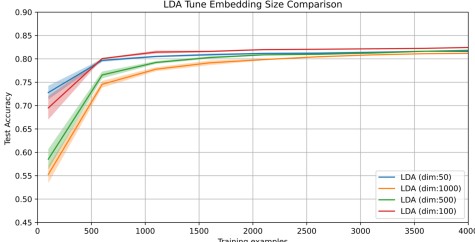

Figure D.1: Test Accuracy for different number of topics used for fitting LDA embedding model on the AG News dataset (Using N=10 for 95% Confidence Interval).

For our own self-supervised method, to extract a representation, we attempted at 1) *softmax+last layer*: apply a Softmax directly to the last layer output, 2) *Word2vec+last layer*: apply an Word2vec embedding matrix to last layer output after Softmax to reduce dimension (the Word2vec matrix is trained over the unsupervised dataset), and 3) *softmax+second2last layer*: apply a Softmax function on top of the second-to-last layer (equivalent to applying an identity matrix to replace the last layer). The dimension of representation with Word2vec is 300 while the softmax + second2 last layer has a dimension of 4096. The original last layer representation has a dimension of the vocabulary size (around 16,700), and has shown inferior results than those with dimension reduction techniques.

| Embedding dimension | Test Accuracy |
|---|---|
| 200 | **0.8498 ± 0.0007** |
| 300 | 0.8488 ± 0.0007 |
| 500 | 0.8484 ± 0.0007 |
| 700 | 0.8483 ± 0.0006 |
| 1000 | 0.8488 ± 0.0009 |
| 2000 | 0.8473 ± 0.0006 |
| 3000 | 0.8481 ± 0.0001 |
| 5000 | 0.8482 ± 0.0006 |

Table D.1: Test Accuracy for different embedding dimension used for fitting Skip-gram word embedding model on the AG News Dataset (Using N=10 for 95% Confidence Interval). We fixed the window-size to be default of 5 and used1 all 4000 training examples.

We included a comparison of the two approaches with dimension reduction in Table D.2, from which we observe that a Softmax on the second-to-last layer output performed better across layers of 3,4 and 5. We reported the result using the *softmax+second2last layer* in Figure 2.

| Test Accuracy | Method | | |
|---|---|---|---|
| # layers (residual) | *Word2vec+last layer* | *softmax+second2last layer* | *softmax+lastlayer* |
| 3 | 0.8508 | **0.8714** | 0.8395 |
| 4 | 0.8450 | **0.8621** | 0.8430 |
| 5 | 0.8492 | **0.8689** | 0.8382 |

Table D.2: Test Accuracy for different representation extraction method. We fixed the rest of hyperparameters to be: 5000 embedding dimension, 150 epochs, 0.0002 learning rate, sampling 4 words in labels, weight decay of 0.01 and resample rate of 2.

## D.3 MODEL ARCHITECTURE

We include here more details on our model architecture used in real data experiments. We used residual model, and tested two main hyperparameters: number of residual blocks and hidden dimension size. We fixed the rest of hyperparameters: we used a 5000 embedding dimension and trained for 150 epochs, with 0.0002 learning rate, weight decay of 0.01 and resample rate of 2. We sampled 4 words in labels for variance reduction purpose.

**Varying Depth and Width** We investigated both the effect of model depth and width on the performance of RBL representation. We trained networks of width 2048, 3000, and 4096 nodes respectively, where the number of node refers to the number of node in the linear layer inside a residual block. We varied the depth of the network by choosing to place 3, 4 and 5 of such block.

As shown in Figure D.2, it appears that using wider models in the unsupervised phase leads to better performance when training a linear classifier on the learned representations. Number of residual block does not seem to have a clear relationship with test accuracy from limited experiments we ran. The best accuracy is achieved when we have 3 residual blocks with dimension of 4096, which is what we used to generate results in Figure 2.

## D.4 MORE REAL-DATA EXPERIMENTS

In addition to the experiments on the AG news dataset, we also run experiments on the IMDB movie review sentiment classification dataset by Maas et al. (2011) and the Small DBpedia ontology dataset, which is a subset of the DBpedia ontology by Zhang et al. (2015), following the preprocessing by Tosh et al. (2021b). In particular, the IMDB movie review dataset contains a unlabeled set of 50k movie reviews, a training set of 12.5k positive movie reviews and 12.5k negative movie reviews, and a test set of 12.5k positive movie reviews and 12.5k negative movie reviews. We randomly selected 1k documents from each class in the training set as our training document and selected 3.5k documents in the test set as our test documents, and used the remaining 91k documents as our unsupervised dataset for training our SSL model to learn document representation. After filtering,

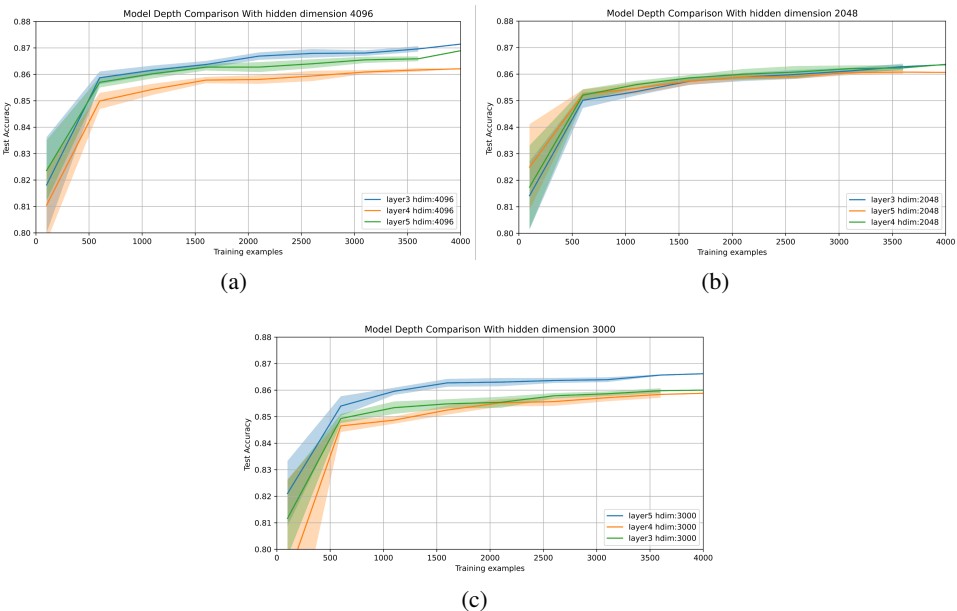

Figure D.2: RBL representation performance varies with different residual model capacity: the best performing is the one with 3 layers and 4096 hidden dimensions, and it was what we used for Figure 2. In all these runs, we use Softmax on second-to-last layer to obtain our representations

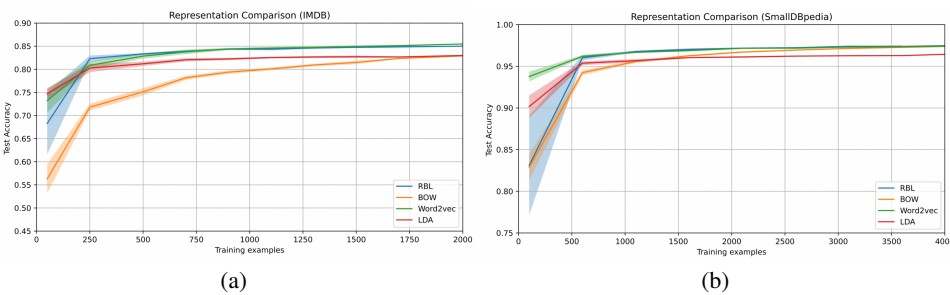

Figure D.3: Performances of RBL and baselines (BOW, Word2vec, LDA) on IMDB and SmallDBpedia datasets. RBL generally outperforms all baselines.

the vocabulary size is about 35k. The Small DBpedia dataset consists of 4 non-overlapping classes, which are company, artist, athlete, and office holder from the original DBpedia ontology dataset, and each class consists 40k training documents and 5k test documents. We randomly chose 1k training documents each class as labeled training documents and used the rest of training documents as the unsupervised dataset, and used the original test documents as our test dataset. After we filtered rare words in the original DBpedia dataset, the vocabulary size is about 10k.

In our experiments, the setup of the baselines follows Section D.2. We plot the test accuracy of our method against the baselines in Figure D.3.We did not include NCE in representation comparison since limited tuning does not achieve comparable results as in Tosh et al. (2021b).Parameter tuning result for LDA and Word2vec is documented in Figure D.4 and Table D.3 respectively.

In the left panel of Figure D.3, we compare the performance of semi-supervised learning on IMDB dataset of our representation RBL as well as baselines. We point out that in general, RBL has similar performance to Word2vec, outperforming LDA and BOW. However,when labeled training data is less abundant, Word2vec and LDA have slightly stronger performance.

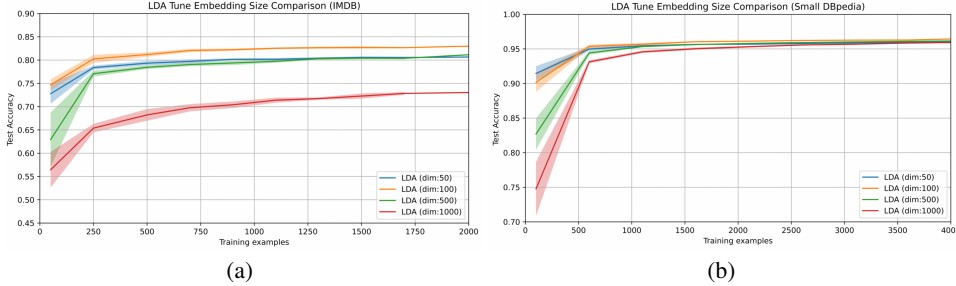

Figure D.4: Test Accuracy for different number of topics used for fitting LDA embedding model on the IMDB and Small DBpedia dataset. We use 100 as final dimension for both datasets based on the tuning above.

| Test Accuracy | Datasets | |
|---|---|---|
| Embedding Dimensions | IMDB | Small DBpedia |
| 100 | 0.8423±0.0014 | 0.9440± 0.0007 |
| 200 | 0.8529±0.0012 | 0.9541±0.0004 |
| 500 | **0.8545±0.0011** | **0.9568±0.0004** |
| 1000 | 0.8544± 0.0005 | 0.9567± 0.0003 |

Table D.3: Test Accuracy for different embedding dimension used for fitting Skip-gram word embedding model on the IMDB and Small DBpedia Dataset (Using N=10 for 95% Confidence Interval). We fixed the window-size to be default of 5 and use all available training examples. We use a dimension of 500 for both datasets for final baseline based on the results above.

In the right panel of Figure D.3, we perform a similar representation comparison on Small DBpedia dataset. While RBL outperforms all other baselines when there are more than around 1000 training examples, it performs worse than Word2vec and LDA when labeled training data is less abundant.

