# OpenReview forum: "Understanding The Robustness of Self-supervised Learning Through Topic Modeling"
_ICLR.cc/2023/Conference — ICLR 2023 poster_

### Official Review · Reviewer_KUB2 · 2022-10-23

**Confidence:** 3
**Correctness:** 4
**Technical Novelty And Significance:** 4
**Empirical Novelty And Significance:** 3
**Recommendation:** 6

**Clarity, Quality, Novelty And Reproducibility:**


## Novelty

The paper cites adequate recent work in this space.
Tosh et al. (2021a) is the most closely related study, which shows "contrastive learning is capable of recovering a representation of documents that reveals their underlying topic posterior information to linear models".

This previous work focuses on contrastive SSL, while the present paper's focus is primarily on reconstruction SSL. The present paper does look at the contrastive setting (see Sec. 4), and the theorem there nicely removes the anchor words assumption needed by Tosh et al.



## Quality

Overall, the study and especially the synthetic experiments seem well designed.

However, there a couple major issues I'd like to hear more about

### W1: Missing assessment of sensitivity to length of document / assymmetric priors

The successful recovery of the posterior despite complete lack of knowledge of the prior is somewhat surprising. I wonder if two things could explain this:

* the likelihood is dominating the per-document posterior in most of the experiments considered, and the specific choice of prior is thus less relevant
* the chosen priors used are fairly default in their assumptions (e.g. LDA uses a symmetric Dir(1/K) prior, so all topics are equally likely to appear in the document)

I'm curious if the results in Sec 5 would hold even if we had an asymmetric prior (e.g. LDA where some topics more common than others) and if test time documents were significantly shorter (e.g. maybe 2-10 words, much smaller than expts in Sec 5 which seem to use Poisson(30) words).

Clearly, in the limit where there are *no* observed words in a test document, the posterior must equl the prior, and the SSL doesn't have access to the prior (though I guess it can try to learn it from abundant training data). I'm just hoping to understand the limits of the proposed method for recovering the posteriors of classic probabilistic models.


### W2: Better to compare predicted distributions over words than topics?

The comparisons in Table 1 all assess estimated per-document distributions over topics, denoted w

I'm wondering if there's a chance that if a (misspecified model) gets this distribution wrong, it could still have a good prediction for the missing word. For example, in a given fixed topic-word matrix A, two different topics can put mass on the same word v. As long as the posterior predictive over words (=Aw) is close to the truth, the model would predict well, even if the model uses a different topic than the "true" one.

Do the authors have thoughts about this?
Probably the topics are distinct enough that there is one clearly "best" w for each document, but I'd just like to check in on this.


## Clarity / Reproducibility

Overall mostly easy to understand, I appreciated that in a paper with theory-heavy contributions the practical implications of the theorems (e.g. Theorem 3.1) were clarified to the reader in the main text.

Major issues are listed here

### W3: Details of NUTS-based posterior sampling missing

Even though widely used in probabilistic programming packages, I've often found NUTS with default settings may not sample well from some distributions without further tuning. Because NUTS isn't commonly used to do posterior inference for models like LDA (even though of course it seems perfectly suitable), I'm hoping to hear a little more about what diagnostics you've performance to be satisfied your chains are converging to the intended posterior distribution.

For example, for LDA have you compared your NUTS results to classic Gibbs samplers for LDA? Or to the "true" w used to generate a document?

Also, when you know the "true" generative model used to simulate the data (as in Table 1), do you really need to use NUTS at all?

### W4: Clarify how topic-word parameters A are handled in experiments

In Table 1 and 2, can you clarify whether A is held fixed to the true topic-word parameters used to generate all data? And that each method is only solving the posterior estimation of w given observed x and fixed A? That detail is missing in the paper.

Furthermore, the text says "Often A will be drawn from ... Dir(\alpha /K)...."

Are there cases where you generate A some other way? If so, how?

### W5: Releasing code would significantly help reproducibility

Hopefully this is an easy one to address.
I think the community would benefit from being able to compare to the toy and real datasets studied here under similar settings.


Minor issues:

* Would be useful to be clear (in supplement) how exactly you calculate the 95% confidence intervals in various tables.


**Strength And Weaknesses:**


## Strengths

* Paper addresses a timely question (understanding the representational power of self-supervised learning)
* Theoretical results cover both reconstruction (Sec 3) and contrastive (Sec 4) paradigms of SSL
* The implications of Theorem 3.1 are reasonably explained to the reader
* Comparisons to many (4) distinct types of topic models in Sec. 5 help back up claims
* Real data demonstration in Sec. 5 looks promising

## Weaknesses

Here I list the primary weaknesses as I see them (each is elaborated below under its respective heading of Quality/Clarity/Reproducibility)

* W1: Missing assessment of sensitivity to length of document / assymmetric priors
* W2: Better to compare predicted distributions over words than topics?
* W3: Details of NUTS-based posterior sampling missing
* W4: Clarify how A is handled in experiments
* W5: Releasing code would significantly help reproducibility



**Summary Of The Paper:**

This paper considers the problem of representation learning for bag-of-words count data, trying to understand the usefulness of recent self-supervised learning, including reconstruction objectives (Eq 1) and contrastive objectives (Eq 2). The key question is: why might these approaches be better than previous ones? The paper's goal is to develop theorems and experiments that try to compare/relate the representations produced by these objectives to the per-document posterior representations of probabilistic topic models (like Latent Dirichlet Allocation).

The main claims see to be that SSL representations are

- as good as the true topic model when the data is generated by a true topic model (backed by theorems in Sec 3/4 and experiments)
- better than a topic model that is *misspecified* for the data (backed by experiments in Sec 5)

Taken together, the paper suggests that SSL is *robust* to model misspecification, and this may help explain why it is a performant choice for real bag-of-words data.

I'll note the first claim seems somewhat surprising, given that a key part of a topic model is its prior over the document's distribution over topics $$w$$, and the SSL does not know about this prior at all, yet recovers moments of the posterior well.

Sec. 5 performs experiments on synthetic data where the true topic model is known (several possible models are considered: pure topic model, LDA, Correlated Topic Model). Here, results in Table 1 assess the distance between the SSL recovered posterior and the "true" posterior (fit with NUTS).

Sec. 6 provides one brief experiment on real data (AG news, where each doc is one of four categories). Here, representations are assessed in terms of their ability to predict the category of documents in the test set. The authors show how reconstruction-based SSL outperforms other baselines as training set size varies from 100-400 documents.

While most theoretical results (Sec. 3) and experiments (Sec. 5-6) focus on reconstruction-based SSL, contrastive SSL is the focus of a theorem in Sec. 4.


**Summary Of The Review:**

Overall, I think this study offers some nice new insights about why self-supervised learning might be effective on data from a wide range of possible models. If my concerns are addressed, I think this could be a valuable paper to accept.

---

> ### Author Response · Authors · 2022-11-11
> **Response**
>
> Dear Reviewer KUB2,
>
> We would like to thank you for the detailed and valuable review. We address your concerns as below.
>
> First, we agree with you that self-supervised learning's success in recovering posterior without knowledge of the prior is indeed surprising (and nice too!) The reason of such success is because self-supervised learning can learn a distribution that is linear in the posterior mean regardless of the prior type (Theorem 3.1). And just like you mentioned, this requires learning from a large amount of training data.
>
>
>
> - W1: Missing assessment of sensitivity to length of document / asymmetric priors
>
>     We believe our experiments already demonstrate that the success of self-supervised learning is not due to the two reasons mentioned in the review (that the likelihood is dominating or the chosen prior is too generic). This is highlighted in the experiments for Correlated Topic Model (CTM) and Pachinko Allocation Model (PAM).
>     In particular, we designed the prior of CTM and PAM to be asymmetric and the topics are correlated with each other. Therefore, in these two settings the prior in fact matters, and self-supervised learning is still able to learn useful representations. In Table C.1 and C.2, we show that using a generic prior such as LDA makes the performance significantly worse compared with SSL representation. The reason of such success is because self-supervised learning learns a distribution linear in the topic posterior mean from abundant training data, even though it does not have access to the prior. We did not explain the setting for the CTM and PAM model clearly in the previous version and we make that more precise in the revision in Appendix C.
>
>
> $~$
>
> - W2: Better to compare predicted distributions over words than topics?
>
>     We would like to first clarify the reason that we choose to measure the performance on the predicted topic proportion instead of word distribution. In this paper, our motivation is to study why the self-supervised learning could learn useful representations, and to define ``useful representation" we focus on the topic model setting and view the topic proportion of each document as its representation. Therefore, to compare the quality of learned representation of different methods, we believe comparing the predicted distributions over topics is a more natural choice than over words.
>
>     Indeed, we believe comparing the predicted distribution over words will also show SSL's advantage, because the topic-word matrix $A$ is randomly generated and well-conditioned (see Figure C.4 in appendix for condition number of $A$). To verify this, we now show the TV distance over word distribution below.

---

> > ### Author Response · Authors · 2022-11-11
> > **Response -continue**
> >
> > Document Type ($\alpha=1$)
> > | Method              | Pure                     | LDA                      | CTM                      | PAM                      |
> > |---------------------|--------------------------|--------------------------|--------------------------|--------------------------|
> > | LDA                 | 0.0361 ± 0.0017          | -                        | **0.0602 ± 0.0043**   | 0.0789 ± 0.0053          |
> > | CTM                 | 0.1909 ± 0.0033          | 0.1658 ± 0.0052          | -                        | 0.1619 ± 0.0043          |
> > | PAM                 | 0.3443 ± 0.0033          | 0.2944 ± 0.0096          | 0.1941 ± 0.0045          | -                        |
> > | **SSL (ours)** | **0.0255 ± 0.0001** | **0.1204 ± 0.0002** | 0.0611 ± 0.0001          | **0.0624 ± 0.0001** |
> >
> > Document Type ($\alpha=3$)
> > | Method              | Pure                     | LDA                      | CTM                      | PAM                      |
> > |---------------------|--------------------------|--------------------------|--------------------------|--------------------------|
> > | LDA                 | 0.0397 ± 0.0017          | -                        | 0.0688 ± 0.0052          | 0.0812 ± 0.0054          |
> > | CTM                 | 0.1883 ± 0.0032          | 0.1493 ± 0.0043          | -                        | 0.1398 ± 0.0034          |
> > | PAM                 | 0.3225 ± 0.0028          | 0.2458 ± 0.0065          | 0.1695 ± 0.0039          | -                        |
> > | **SSL (ours)** | **0.0319 ± 0.0001** | **0.0987 ± 0.0002** | **0.0659 ± 0.0001** | **0.0616 ± 0.0001** |
> >
> > Document Type ($\alpha=5$)
> > | Method              | Pure                     | LDA                      | CTM                      | PAM                      |
> > |---------------------|--------------------------|--------------------------|--------------------------|--------------------------|
> > | LDA                 | 0.0461 ± 0.0028          | -                        | **0.0614 ± 0.0045** | 0.0796 ± 0.0051          |
> > | CTM                 | 0.1885 ± 0.0037          | 0.1402 ± 0.0035          | -                        | 0.1282 ± 0.0035          |
> > | PAM                 | 0.3078 ± 0.0031          | 0.2171 ± 0.0050          | 0.1577 ± 0.0036          | -                        |
> > | **SSL (ours)** | **0.0367 ± 0.0004** | **0.0919 ± 0.0002** | 0.0624 ± 0.0001          | **0.0571 ± 0.0001** |
> >
> > Document Type ($\alpha=7$)
> > | Method              | Pure                     | LDA                      | CTM                      | PAM                      |
> > |---------------------|--------------------------|--------------------------|--------------------------|--------------------------|
> > | LDA                 | 0.0483 ± 0.0024          | -                        | 0.0656 ± 0.0047          | 0.0822 ± 0.0057          |
> > | CTM                 | 0.1863 ± 0.0036          | 0.1304 ± 0.0033          | -                        | 0.1259 ± 0.0040          |
> > | PAM                 | 0.2927 ± 0.0026          | 0.1990 ± 0.0048          | 0.1492 ± 0.0036          | -                        |
> > | **SSL (ours)** | **0.0348 ± 0.0001** | **0.0940 ± 0.0002** | **0.0623 ± 0.0001** | **0.0549 ± 0.0001** |
> >
> > Document Type ($\alpha=9$)
> > | Method              | Pure                     | LDA                      | CTM                      | PAM                      |
> > |---------------------|--------------------------|--------------------------|--------------------------|--------------------------|
> > | LDA                 | 0.0570 ± 0.0036          | -                        | 0.0644 ± 0.0047          | 0.0826 ± 0.0054          |
> > | CTM                 | 0.1876 ± 0.0040          | 0.1265 ± 0.0032          | -                        | 0.1218 ± 0.0037          |
> > | PAM                 | 0.2848 ± 0.0030          | 0.1889 ± 0.0044          | 0.1423 ± 0.0031          | -                        |
> > | **SSL (ours)** | **0.0386 ± 0.0004** | **0.0866 ± 0.0002** | **0.0566 ± 0.0001** | **0.0522 ± 0.0001** |

---

> > > ### Author Response · Authors · 2022-11-11
> > > **Response -continue**
> > >
> > > - W3: Details of NUTS-based posterior sampling missing
> > >
> > >     (1) Check the convergence of NUTS
> > >
> > >     In our experiments, we use PyMC3 package to perform NUTS-based MCMC posterior sampling. Specifically, we use 3000 tuning iterations to tune the sampler's step size so that we approximate an acceptance rate of 0.9. After tuning, we draw 2000 samples from the posterior distribution and use the mean of these 2000 samples as our best estimate for the topic posterior mean.
> > >
> > >     In Figure C.5 in the appendix, we show a few examples of comparing different posterior with the ground truth topic proportion (the documents are generated by PAM). We can see that the PAM MCMC posterior (top right in Figure C.5) is indeed similar to the ground truth $w$ used to generate the document (bottom right in Figure C.5), which suggests that the NUTS is effective. For LDA documents, as shown in Figure C.6, we also observe that MCMC posterior assuming LDA model (bottom row in Figure C.6) is also very similar to the ground truth $w$ (top row in Figure C.6) used to generate the documents.
> > >
> > >     To further validate our NUTS-based MCMC topic posteriors, we now run Gibbs sampling for the LDA documents and compare the results. We find that the topic posterior obtained from NUTS is indeed very similar to the topic posterior from Gibbs sampling, as the average TV distance between them (averaged over all LDA test documents) is very small (shown below). We've updated our code to include the LDA Gibbs sampling.
> > >
> > >     | $\alpha$              | 1                     | 3                     | 5                     | 7                     | 9                     |
> > >     |-------------------|------------------------|------------------------|------------------------|------------------------|------------------------|
> > >     | TV                 | 0.0159          | 0.0178          | 0.0231          | 0.0245          | 0.0269          |
> > >
> > >
> > >     $~$
> > >
> > >     (2) Necessity of using NUTS when knowing the ground truth.
> > >
> > >     Though we know the ground truth topic proportion $w$ of each document, the best estimation one could have after seeing the document alone is in fact the posterior of topic proportion under the correct prior. However, such posterior in general does not have closed-form expression and we need to use sampling method to estimate the posterior.
> > >
> > > $~$
> > >
> > > - W4: Clarify how topic-word parameters A are handled in experiments
> > >
> > >     In our experiments, the topic-word matrix $A$ is known and fixed, and we estimate the posterior given document $x$ and topic-word matrix $A$. For pure topic and LDA experiments, each column of $A$ (i.e., each topic's word distribution) is generated from $Dir(\alpha/K)$. For CTM and PAM, our goal is to construct $K/4$ groups of 4 topics such that within each group, topic 0 and topic 1 have similar word distribution, and topic 2 and topic 3 have different word distribution (and different from the word distribution of topic 0 and 1 as well). To achieve this, we first sample $K/4$ word distribution vectors from $Dir(\alpha/K)$, and for each word distribution vector, we permute its entries to get the word distribution of topic 0,1,2,3 within each group and intentionally align the large entries in topic 0 and 1 to ensure high similarity between them. We clarify this process in the revision in Appendix C.
> > >
> > > $~$
> > >
> > > - W5: Releasing code would significantly help reproducibility
> > >
> > >     The code for the experiments is provided in the supplement materials.
> > >
> > > $~$
> > >
> > > - The confidence interval reported in the paper is calculated as $mean$ ± $ 1.96*s_{test docs}/\sqrt{N_{test docs}} $, where $mean$ is the mean of TV distance (or accuracy), $s_{test docs}$ is the standard deviation of TV distance (or accuracy) and $N_{testdocs}$ is the number of test documents. We add these details in the revision in Appendix C.

---

> > > > ### Comment · Reviewer_KUB2 · 2022-11-19
> > > > **Consider W3, W4, and W5 resolved**
> > > >
> > > > I appreciate the authors' response to these points above. Thank you for your detailed reply.
> > > >
> > > > I don't have concerns about these items. I hope the revised manuscript makes each of these crystal clear.

---

> > ### Comment · Reviewer_KUB2 · 2022-11-19
> > **Reply to authors on W1 and W2**
> >
> > Thanks to the authors for their carefully thought out comments.
> >
> > RE W1: I agree the provided CTM experiments in Table C1 suggest that the proposed SSL approach seems to handle asymmetric priors.
> >
> > RE W2: OK, if I compare the TV in topics (Table C1) to the tables below, and focus on cases where CTM is true model, I guess I see an interesting pattern:
> >
> > * TV on topics seems to favor the SSL approach over plain LDA decisively (TV of SSL is ~0.05-0.14 lower than TV of LDA)
> > * TV on words is less clear, for CTM column SSL and LDA both get TV of around 0.06 in all alpha settings
> >
> > From this, I don't think we can conclude that "predicted distribution over words will yield similar results to comparing the predicted distribution over topics"
> >
> > I hope you can revise accordingly. This probably needs some more discussion

---

> > > ### Author Response · Authors · 2022-11-22
> > > **Further Response on W2**
> > >
> > > We appreciate your feedback and we are glad that our response addressed your concerns on W1, W3, W4, and W5. For W2, when we wrote "predicted distribution over words will yield similar results to comparing the predicted distribution over topics", what we meant by "similar" was that we expected SSL would outperform misspecified model on word distribution in most cases (note that there are a few cases where the mean for LDA on CTM is lower, but they also have large variance so their confidence intervals overlap; on the other hand for the largest $\alpha$ the SSL is better and the confidence intervals do not overlap). We thank the reviewer for pointing this out, and we'll more carefully rephrase the sentence in our earlier response to avoid confusion.
> > >
> > > There is a clear reason why for the documents generated by CTM models the gap becomes smaller. This is because the topic-word matrix $A$ for CTM is intentionally constructed so that some pairs of topics have similar word distributions, and thus different topic proportion vectors can lead to similar word distributions, while CTM will be able to disambiguate these pairs of topics by leveraging the correlation structure in the prior. To see this more clearly, let's consider the following toy example:
> > >
> > > Assume we have $K=4$ topics (denoted as topic 0, 1, 2, and 3) and each topic's corresponding word distribution (denoted as $v_0$, $v_1$, $v_2$, $v_3 \in \mathbb{R}^V$), where $v_0$ and $v_1$ are very similar (TV distance between $v_0$ and $v_1$ is between 0.23 and 0.27, and their cosine similarity is above 0.8, as in our experiments), topic 0 and 2 are correlated, and topic 1 and 3 are correlated. Note that this setup is exactly the same as our CTM experiments described in Section 5.1, except that here we are using $K=4$ instead of $K=20$ for simplicity. Now consider a CTM document $D$ generated from a ground-truth topic proportion $w_{true} = (0.6, 0, 0.4, 0)$, the word distribution in $D$ will follow the distribution $Categorical(0.6 v_0 + 0.4 v_2)$, which is very similar to $Categorical(0.6 v_1 + 0.4 v_2)$. Since LDA model does not know the correlation between topics, it cannot distinguish whether $D$ is generated from topic proportion $w_{true} = (0.6, 0, 0.4, 0)$ or $w_{wrong} = (0, 0.6, 0.4, 0)$, or even $w_{wrong}' = (0.3, 0.3, 0.4, 0)$, and we can expect that sometimes the topic proportion recovered by LDA model is a *wrong topic proportion that gives the approximately correct word distribution*. Therefore, although LDA can recover the word distribution reasonably well, it cannot reliably recover the true topic proportion. On the other hand, SSL is robust and can learn the correlation between common words in topic 0 and 2, and when it sees common words in topic 2 and no words from topic 3 in document $D$, it can infer that topic 0, instead of topic 1, is salient in the topic proportions vector.
> > >
> > > We hope this toy example can shed some light on why SSL and LDA have similar TV on the word distribution in CTM documents even though SSL's recovered topic proportion is better. Since the recovery of topic proportions is of primary interest in our paper, we believe SSL's superiority over misspecified topic model is already demonstrated by the comparison of their TV on the distribution over topics.
> > >
> > > Please let us know if there is anything else we can clarify on!

---

### Official Review · Reviewer_6Fra · 2022-10-23

**Confidence:** 3
**Correctness:** 4
**Technical Novelty And Significance:** 3
**Empirical Novelty And Significance:** 3
**Recommendation:** 6

**Clarity, Quality, Novelty And Reproducibility:**

Overall this paper is clearly written. However, the novelty is not very clearly mentioned. Showing the details of existing work [Tosh et al.] and clarifying the extension would be necessary.

**Strength And Weaknesses:**

Strength
- This paper tackles an important problem. Numerous theoretical analysis on SSL has been published in recent years. Most of them focus on the generic effectiveness of the feature representation of the SSL for downstream tasks. This paper restricts the problem to the topic model that includes the main applications of SSL. The robustness analysis in Theorem 3.3 is practically important.

Weaknesses
- This paper includes a generalized result of [Tosh et al.'21]. However, a detailed comparison with [Tosh, et al.'21] is not presented. More concretely, which condition is relaxed compared to [Tosh et al.'21]? How much is the practical usefulness of such a relaxation?
- The required dimension of the feature representation is relatively high. Is it possible to show how to reduce the dimension of the functions f and g to obtain similar theoretical results?


**Summary Of The Paper:**

This paper studies self-supervised learning (SSL) for general topic models. The main approaches of SSL include reconstruction-based objective and contrastive objective. The paper investigates that feature representations given by SSL provide basis functions of the higher order posterior mean. Also, the robustness analysis provides the error bound of near-optimal solutions in the reconstruction-based SSL. In numerical experiments, the reconstruction-based SSL is examined and compared to the inference with some topic models. The numerical results indicate that the SSL approach is superior to the statistical inference using misspecified models.

**Summary Of The Review:**

This paper tackles an important problem. However, the novelty compared to existing works is not clearly discussed.

---

> ### Author Response · Authors · 2022-11-11
> **Response**
>
> Dear Reviewer 6Fra,
>
> We appreciate your valuable feedback and would like to address the concerns as below:
>
> - Comparison with [Tosh et al.'21]
>
>     We provide a more detailed comparison between our results with [Tosh et al.'21] as below: (1) We consider both reconstruction-based objective and contrastive objective (two commonly used objectives in self-supervised learning), while [Tosh et al.'21] focused only on the contrastive objective; (2) For the results of contrastive objective, we are able to remove the anchor word assumption used in Theorem 3 in [Tosh et al.'21], which states that for each topic there is at least one word that cannot possibly appear in any other topic. Such assumption is restrictive and may not always hold in practice, and we show that self-supervised learning can still learn useful representation when this assumption is removed. The comparison has been included in the related work section.
>
> - The dimension of feature representation
>
>     For contrastive objective, the dimension of feature representation is $K^t$, which is relatively small because the number of topics $K$ is usually small. For reconstruction-based objective, although the dimension of $f(x)$ is $V^t$, its ``effective dimension" is actually $K^t$, which is much smaller than $V^t$. To see this, let's consider the simple case of $t=1$. As we discussed in the paragraph below Theorem 3.1, we have $f(x)=A\mathbb{E}[w|x,A]$, where $A\in \mathbb{R}^{V \times K}$ is the topic-word matrix and $\mathbb{E}[w|x,A]$ is the posterior mean of $w\in\mathbb{R}^K$. In words, this means that $f(x)$ is completely determined by a linear transformation of a $K$-dimensional vector. Therefore, when we consider the feature of $n$ documents as a matrix $[f(x^{(1)}),...,f(x^{(n)})]\in\mathbb{R}^{V\times n}$, the rank of this matrix is at most $K$ and one can get these $K$-dimensional features by running standard dimension reduction method (e.g., SVD) on the original feature matrix. In the revision, we add a paragraph below Theorem 3.1 and add the above discussions in Appendix A.4.

---

### Official Review · Reviewer_DvbM · 2022-10-24

**Confidence:** 3
**Correctness:** 3
**Technical Novelty And Significance:** 4
**Empirical Novelty And Significance:** 3
**Recommendation:** 6

**Clarity, Quality, Novelty And Reproducibility:**

The paper lacks a bit of clarity in eventually linking the polynomial form of w with the linear classification error. This helps to verify the significance of the proposed methods.

The paper is novel in the sense that this is the first paper I came across in attributing the model mis-specification with the actual performance of SSL methods.

**Strength And Weaknesses:**

Strength:

The paper demonstrates the ability of self-supervised learning to adapt to different models. Since the prior distribution is not present in the SSL models,  SSL methods therefore avoid the risk mis-specifying the model. This helps the model to better inference the posterior of the topics. In the meanwhile, the paper gives the explicit theoretical guarantees for self-supervised learning with the topic modeling task,  and showed that SSL can provide useful information about the topic posterior without bounded error, even if SSL does not have prior information about the topics. This observation is relatively new and fresh to me.

Weakness:

The paper is a bit hard to follow in the sense that the proposed theories are very loosely connected to the actual classification error on the downstream task in the form of the polynomials of the w variables. I would appreciate it if the authors might comment on the significance of assuming error bound on the polynomials form of the posterior on w variables.


**Summary Of The Paper:**

This paper demonstrates the point that Self-supervised learning is immune to the choice of the probabilistic model, therefore shows robustness to model misspecification. In contrast, traditional approach of probabilistic modeling hinges on the specific predefined model. Therefore, the advantage of self-supervised learning is that SSL may outperforms traditional model when inferencing with misspecified model. The evaluation parts measures topic posterior recovery loss as the Total Variation (TV) distance between the recovered topic posterior and the ground truth topic posterior mean, supporting the advantage of the SSL methods.

**Summary Of The Review:**

The paper gives a series of theorems justifying the bounded error of topic classification task, when using the features out of the SSL model.  The paper is well written, and evaluation supports the claims. I have some concerns in linking the theorems to the actual classification error of the topics, I expect to hear from the authors in this regard.

---

> ### Author Response · Authors · 2022-11-11
> **Response**
>
> Dear Reviewer DvbM,
>
> We appreciate your valuable comment and address the concerns below:
>
> - Linking the polynomial form of $w$ with the linear classification error.
>
>     Note that the set of polynomial functions of $w$ is able to approximately represent a large set of functions (e.g., the continuous functions). Indeed, as mentioned in the [Tosh et al.'21] (see paragraph below Theorem 3 in their paper), "with the Stone-Weierstrass theorem, in principle, the posterior mean of any continuous function of $w$ can be approximated using our representation (polynomial of $w$)." Therefore, if the target function of downstream task is a continuous function of $w$, then we should expect to achieve low classification error when using the representations learned by self-supervised learning.

---

> > ### Comment · Reviewer_DvbM · 2022-11-25
> > **Thanks for your response**
> >
> > Thanks for your response, I appreciate your answers and I maintain my score.

---

### Decision · Program_Chairs · 2023-01-20

**Decision:**

Accept: poster

**Justification For Why Not Higher Score:**

The improvement over SOTA is not that spectacular. However this is to be expected as the field of Topic models is more than two decades old. Also, the referees did not raise the scores after perusing the rebuttal. This makes it difficult to make a stronger case.

**Justification For Why Not Lower Score:**

Most issues raised by the reviewers have been addressed by the authors.

**Metareview: Summary, Strengths And Weaknesses:**

This paper explores the quality of representations learnt by Self-supervised learning(SSL) through Topic models. It argues that Topic models are robust to model mis-specification when learnt either through contrastive or reconstruction based objective functions.
The main result
is the paper shows that the obtained representations can represent any polynomial function of the posterior weights given enough data. The line of investigation is similar to earlier work by Tosh( et al'21). However they go beyond Tosh et al., by removing the Anchor Word assumption.

The main strengths of the paper are
1. Contributes to the understanding of feature representation through SSL.
2. It provides rigorous statements of the key claims especially the robustness claims

The main weakness of the paper
1. Most of the groundwork is already there in Tosh et al.'21, in particular theorem 3 of the reference. Here the results are extended to Topic models. Here the reasoning remains more or less the same, but the anchor word assumption is relaxed. Appears to be modest contribution wrt to state of the art
2. The technical presentation can significantly improve







**Note From Pc:**

if the above contains the word "oral" or "spotlight" please see: "oral" presentation means -> notable-top-5% and "spotlight" means -> notable-top-25%. As stated in our emails, we are disassociating presentation type from AC recommendations

**Summary Of Ac-Reviewer Meeting:**

Unfortunately we did not have a meeting yet.
The scores stand at 6 and are unchanged. I requested the referees to consider revising the scores as per their assessment.
I am yet to hear from them. Was planning to have the meeting after i hear from them.